EMBO
Molecular Medicine

# Novel AAV capsids for intravitreal gene therapy of photoreceptor disorders

Marina Pavlou[1,2], Christian Schön[2], Laurence M Occelli[3], Axel Rossi[4], Nadja Meumann[4,5], Ryan F Boyd[6], Joshua T Bartoe[6], Jakob Siedlecki[1], Maximilian J Gerhardt[1], Sabrina Babutzka[1,2], Jacqueline Bogedein[1,2], Johanna E Wagner[2], Siegfried G Priglinger[1], Martin Biel[2], Simon M Petersen-Jones[3], Hildegard Büning[4,5,*] (iD) & Stylianos Michalakis[1,2,**] (iD)

## Abstract

Gene therapy using recombinant adeno-associated virus (rAAV) vectors to treat blinding retinal dystrophies has become clinical reality. Therapeutically impactful targeting of photoreceptors still relies on subretinal vector delivery, which detaches the retina and harbours substantial risks of collateral damage, often without achieving widespread photoreceptor transduction. Herein, we report the development of novel engineered rAAV vectors that enable efficient targeting of photoreceptors via less invasive intravitreal administration. A unique *in vivo* selection procedure was performed, where an AAV2-based peptide-display library was intravenously administered in mice, followed by isolation of vector DNA from target cells after only 24 h. This stringent selection yielded novel vectors, termed AAV2.GL and AAV2.NN, which mediate widespread and high-level retinal transduction after intravitreal injection in mice, dogs and non-human primates. Importantly, both vectors efficiently transduce photoreceptors in human retinal explant cultures. As proof-of-concept, intravitreal *Cnga3* delivery using AAV2.GL lead to cone-specific expression of Cnga3 protein and rescued photopic cone responses in the *Cnga3*[−/−] mouse model of achromatopsia. These novel rAAV vectors expand the clinical applicability of gene therapy for blinding human retinal dystrophies.

**Keywords** achromatopsia; intravitreal delivery; novel AAV; retina
**Subject Categories** Genetics, Gene Therapy & Genetic Disease; Methods & Resources

## Introduction

As the number of gene therapies obtaining market approval is increasing, focus is now set on improving first-generation delivery tools in terms of efficiency and safety. This includes the recombinant adeno-associated virus (rAAV) vectors that are considered the standard gene delivery tool for gene therapy of acquired or inherited retinal dystrophies (IRDs) (Ali *et al*, 2017). AAV is a *Dependoparvovirus* with no assigned pathogenicity in humans (Berns & Muzyczka, 2017). rAAV vectors lack an integrase activity, transduce proliferating as well as post-mitotic cells, and mediate long-term gene expression from vector episomes in post-mitotic or slowly proliferating cells (Berns & Muzyczka, 2017). It is therefore ideal for pathologies manifesting in post-mitotic neurons which are incapable of innate regeneration. Many IRDs are caused by genetic mutations leading to degeneration of the light-sensing photoreceptors. Consequently, gene therapies targeting photoreceptors or the supporting retinal-pigmented epithelium (RPE) are developed to restore photoreceptor function and inhibit or delay the pathological process. rAAV vectors are extensively used in preclinical and clinical studies for this purpose, with an excellent safety record both for ocular and non-ocular gene therapy (Trapani & Auricchio, 2018). With regard to the eye, only a transient immune response to intraocular vector delivery has been observed so far, which did not limit treatment success (Reichel *et al*, 2018; Rabinowitz *et al*, 2019). Indeed, the efficiency of first-generation rAAV vectors in supplementing the *RPE65* gene function resulted in market authorization of voretigene neparvovec for the treatment of *RPE65*-linked retinal dystrophies, first in the United States and the following year in Europe (Keeler & Flotte, 2019).

In order to target photoreceptors in the outer retina, the standard clinical administration route so far is subretinal, which entails surgical detachment of the retina from the RPE and injection of the vector

1 Department of Ophthalmology, Ludwig-Maximilians-University, Munich, Germany
2 Centre for Integrated Protein Science Munich (CIPSM) at the Department of Pharmacy, Ludwig-Maximilians-University, Munich, Germany
3 Department of Small Animal Clinical Sciences, Michigan State University, East Lansing, MI, USA
4 Laboratory for Infection Biology and Gene Transfer, Institute of Experimental Haematology, Hannover Medical School, Hannover, Germany
5 REBIRTH Research Centre for Translational Regenerative Medicine, Hannover Medical School, Hannover, Germany
6 Ophthalmology Services, Charles River Laboratories, Mattawan, MI, USA
*Corresponding author. Tel: +49 5115325106; E-mail: buening.hildegard@mh-hannover.de
**Corresponding author. Tel: +49 89440053083; E-mail: michalakis@lmu.de

into a temporally formed cavity, i.e. subretinal bleb. This holds true for voretigene neparvovec, which uses AAV serotype 2 (AAV2) and is administered via a one-off subretinal injection in affected individuals. Despite its efficacy in targeting photoreceptors, the subretinal injection procedure can be deleterious to an already compromised retina. With the exception of certain engineered capsids (Khabou *et al*, 2018; Boye *et al*, 2020), conventional AAV serotypes are unable to spread laterally and achieve only local transduction, *i.e.* cells outside the subretinal bleb area are not exposed to sufficient amounts of the vector. As such, there is an unmet need for novel rAAV vectors that can target larger areas of the retina and be applicable via a less invasive route, such as intravitreal injection.

In recent years, various capsid engineering strategies have been employed (Grimm & Büning, 2017) to develop rAAV vectors for enhanced retinal transduction (Petrs-Silva *et al*, 2009; Dalkara *et al*, 2013; Zinn *et al*, 2015; Katada *et al*, 2019). In this study, we combined systemic administration of the AAV library with local recovery of AAV genomes from retinal target cells in order to achieve a very high selection pressure. Three screening rounds were performed *in vivo*, in which the library was administered intravenously in *C57BL6/J* mice and AAV genomes were recovered from retinal cells. To further increase the selection pressure, we limited the incubation time to 24 h. Despite this short incubation period and the challenging administration route, library genomes were detectable in nuclei from retinal cells. This unique and stringent selection procedure yielded AAV2.GL and AAV2.NN, two novel engineered capsid variants, capable of widespread retinal transduction across species after a single intravitreal injection. The novel rAAV vectors can be produced at high yield, are less sensitive to neutralising human serum, and efficiently transduce photoreceptors after intravitreal injections in mice, dogs and non-human primates (NHP), and vitreal administration on human retinal explants. Using the *Cnga3*$^{-/-}$ mouse model of achromatopsia (Biel *et al*, 1999), we also report a first validation of AAV2.GL in a proof-of-concept study for restoring cone photoreceptor function after intravitreal gene supplementation therapy.

## Results

### Library screening in mice

Our goal was to develop capsid variants with an enhanced transduction profile for photoreceptors, capable of crossing biological barriers that limit vector penetrance in the retina, such as the synaptic layers of the retina and the extracellular matrix. We employed a random peptide-display library using the AAV2 capsid as scaffold (Fig EV1A), where a random 7-mer amino acid sequence was inserted between N587 and R588 (Fig EV1B). The resulting library with approximately 5 million unique variants (Perabo *et al*, 2003, 2006b) was counter-selected for wildtype or enhanced heparan sulphate proteoglycan (HSPG) binding using heparin affinity chromatography and then injected intravenously into adult *C57BL6/J* mice (2-months old) with > 8E10 total viral genomes (vg) of the initial library (library #1) per mouse. At only 24h after injection, animals were sacrificed, their retinas harvested, and DNA was isolated. DNA was examined via qPCR for the presence of viral genomes revealing successful accumulation in retina and thus

crossing of extracellular barriers. Subsequently, viral genomes were amplified and served as template for sublibrary production (library #2). This procedure was repeated in the second selection round with intravenous injection of > 5E11 vg of the first sublibrary (library #2). This time, also DNA from nuclei of retinal cells, was isolated and analysed revealing that capsid variants not only reached our target cells and were internalised, but also successfully transported into the nucleus. The second sublibrary (library #3) was generated as before, and in the third selection round, 5E11 vg of library #3 were injected intravenously in both adult *C57BL6/J* and *RG-eGFP* mice (2-months old). 24h after injection, rod photoreceptors were isolated by magnetic-activated cell sorting (MACS) and cone photoreceptors by fluorescence-activated cell sorting (FACS). DNA from AAV variants which had reached rods and/or cones were extracted, and NGS was performed to identify the capsid variants which had most efficiently targeted rods and cones. In addition, we analysed whole retina DNA using samples of selection round #2.

### Side-by-side comparison of AAV candidates

Valid NGS reads from whole retina, rods and cones were compared, where valid means only those containing the sequence encoding for RGNAAA $X_1$ $X_2$ $X_3$ $X_4$ $X_5$ $X_6$ $X_7$ AARQ with "X" indicating the random amino acids (Fig EV1B). Sequences detected in rods and cones but not in the whole retina were excluded. Furthermore, the ratio of valid reads in cones or rods over the whole retina was calculated, to select for those enriched in photoreceptors, favouring the cones. The top five most frequent capsid variants were AAV2.GL, AAV2.NN, AAV2.GA, AAV2.NS and AAV2.SS (Table 1), which were then vectorised and tested *in vivo* in mice. The vectors displayed the selected peptides and encoded a self-complementary (sc) AAV vector genome encoding for enhanced green fluorescent protein (eGFP) as marker gene controlled by the ubiquitous cytomegalovirus (CMV) promoter. To interrogate their transduction capacity in the retina, 2E9 vg of the 5 vectorised candidates were injected intravitreally in 2-month-old *C57BL6/J* mice (*n* = 2–4). At 2 weeks

**Table 1. Capsid insertion amino acid sequence of candidates identified via NGS screening.**

| Abbreviation | Capsid variant insertion sequence | Enrichment factor rods/ whole retina | Enrichment factor cones/ whole retina |
|---|---|---|---|
| AAV2.GL | RGNAAA GLSPPTR AARQ | n/d | 36 |
| AAV2.NN | RGNAAA NNPTPSR AARQ | 137 | n/d |
| AAV2.GA | RGNAAA GAHRSDS AARQ | n/d | 28 |
| AAV2.NS | RGNAAA NSRPAAA AARQ | n/d | 4 |
| AAV2.SS | RGNAAA SSPGLPR AARQ | n/d | 36 |

Sequence is displayed as follows: RGN587 (AAV2 sequence)—AAA (linker sequence)—7mer insertion—AA (linker sequence)—RQ (AAV2 sequence). The enrichment factor was calculated by the ratio of NGS reads in rods or cones versus the whole retina. For variants only found in the rods or cones, this enrichment factor is n/d = not detectable in the corresponding cell population.

post-injection (WPI), *in vivo* cSLO imaging was performed to document eGFP expression (Fig EV2A) which revealed successful transduction by all five capsid variants in the mouse retina. Using constant detector gain to observe comparable levels of fluorescence in the cSLO, it was shown that AAV2.GL, AAV2.NN and AAV2.SS achieved the brightest eGFP fluorescence by 2 WPI. The subsequent histological analysis of retinal cryosections at 3 WPI (Fig EV2B) verified AAV2.GL and AAV2.NN as optimal candidates for further analysis since these capsid variants achieved the highest eGFP expression in the mouse retina. To further confirm our decision, we revised the NGS data and screened for related sequences that were selected in parallel. We found GGSPPYR and SLGGSPR as most closely related sequences to AAV2.GL (Fig EV2C), while for AAV2.NN comparison of related sequences might hint to a PTPS/LR receptor-binding motif (Fig EV2D).

## Novel vectors compared with parental AAV2 and AAV2.7m8 in mice

To evaluate more comprehensively the performance of the novel capsids in the retina, AAV2.GL(GL) and AAV2.NN(NN) were compared with the parental AAV2 and to AAV2.7m8(7m8) (Dalkara *et al*, 2013) as the state-of-art reference. A sc-CMV-eGFP expression cassette was packaged, and a total of 2E9 vg were injected intravitreally in 2-month-old wild-type mice ($n$ = 3–4). The mice were followed up weekly using cSLO imaging of eGFP fluorescence. Animals treated with AAV2 achieved a scattered and weak eGFP expression by 1 WPI, which became gradually more intense at 2 WPI. In contrast, AAV2.7m8-, AAV2.GL- and AAV2.NN-treated eyes had a strong eGFP fluorescence already by 1 WPI, which expanded to most of the mouse fundus by 3 WPI (Fig 1A). At 3 WPI, all mice were euthanized, and retinas were processed for IHC. Confocal scans of retinal cross sections revealed that AAV2-treated eyes had only limited eGFP signal mainly at the ganglion cell layer (GCL), with only sparse eGFP-positive cells in the inner (INL) or outer nuclear layer (ONL). Engineered capsids AAV2.7m8, AAV2.GL and AAV2.NN, however, achieved a strong eGFP signal which spanned throughout all retinal layers (Fig 1B). Notably, eGFP expression from AAV2.GL and AAV2.NN was stronger in a larger portion of the mouse retina compared with AAV2.7m8 (Fig 1C) and co-localised with the photoreceptor-specific marker recoverin (Fig 1D and E). Furthermore, qPCR analysis of eGFP expression from mouse retinal samples ($n \geq 3$) revealed a 13.4-fold increase ($P = 0.0086$) in

transcript levels for AAV2.GL and 12.1-fold increase ($P = 0.0211$) for AAV2.NN compared with AAV2. While AAV2.7m8 had a 5.5-fold increase in eGFP transcripts, this was not significantly different to the AAV2 ($P = 0.2295$) (Fig 1F). To compare the photoreceptor-specific transduction efficiency among capsids, the previous study was repeated and 8E9 vg of different rAAV vectors carrying a single stranded (ss) eGFP cassette under the control of a human rhodopsin (hRho) promoter were intravitreally injected. The fundus eGFP signal was again monitored via in-life cSLO over a 3 week period (Fig 2A). All three engineered AAV variants outperformed AAV2, with AAV2.NN showing the strongest signal. Specifically, AAV2.NN produced a faster and brighter eGFP signal which was already present at 1 WPI, in line with the rod-favouring nature of this novel capsid (Table 1). qPCR quantification at 3 WPI confirmed this finding and showed a significantly higher level of eGFP transcripts in retinas injected with AAV2.NN compared with AAV2 ($P = 0.0002$), AAV2.7m8 ($P = 0.0018$) and AAV2.GL ($0.0313$). AAV2.GL also achieved significantly higher transcript levels than AAV2 ($P = 0.0427$) but only a trending increase compared with AAV2.7m8 ($P = 0.2725$). A trending increase in eGFP transcripts was also recorded in AAV2.7m8-treated retinas compared with AAV2 ($P = 0.2725$) (Fig 2B). Confocal scans of retinal cross sections further confirmed the superiority of AAV2.NN over AAV2 and AAV2.7m8 in targeting photoreceptors in the ONL, while AAV2.GL performed at least as well as AAV2.7m8 (Fig 2C).

## Characterisation of AAV2.GL and AAV2.NN

Given the superior performance of AAV2.GL and AAV2.NN in mouse, we further characterised the novel capsids with respect to yield, physical properties and host-vector interactions. Multiple independent production cycles showed that both AAV2.GL and AAV2.NN produce to the same titre as wild-type AAV2, whether harvested from cell supernatant (Fig EV3A) or cell pellet (Fig EV3B). As previously mentioned, we counter-selected the library for variants with wild-type or stronger heparin affinity (analogue for HSPG) using standard heparin affinity chromatography. Interestingly however, position 7 of the unique peptide insert for both AAV2.GL and AAV2.NN has an arginine residue. This arginine residue, together with the two alanine linker residues and with R588 of the parental AAV2 sequence, forms a potential HSPG-binding motif (Perabo *et al*, 2006a,b). To determine whether the novel variants differ from AAV2 regarding their ability to bind HSPG, a

---

**Figure 1.  AAV2.GL and AAV2.NN achieve panretinal transduction in mice using ubiquitous promoter.**

Vector mediated eGFP expression after intravitreal injection of novel capsid variants compared with controls in 2-month-old mice.

A    *In vivo* cSLO examinations of the mouse retina at 1- and 2-week post-injection (WPI) of 2E9 total vg (in 1 µl) of sc-CMV-eGFP packaged with parental serotype AAV2, state-of-art AAV2.7m8, novel AAV2.GL and AAV2.NN capsid variants.

B    Confocal scans of retinal cross sections immunolabelled for eGFP and DAPI. Acquisition settings were kept constant for all samples. Scale bar: 100 µm.

C    Confocal tile-scans of retinal cross sections immunolabelled for eGFP and Hoechst. Scale bar: 500 µm.

D, E  Immunolabelled cross sections of mouse retina infected with AAV2.GL and AAV2.NN, respectively, with focus on the ONL and INL, stained for eGFP, recoverin and DAPI. Scale bar: 50 µm.

F    qRT–PCR data comparing fold change in eGFP transcript levels from retinal samples infected with capsid variants AAV2.7m8, AAV2.GL and AAV2.NN compared with AAV2. Biological replicates $n \geq 3$. Bars show mean ± SEM. One-way ANOVA was performed with post hoc Holm–Sidak multiple comparisons test. *$P \leq 0.05$, **$P \leq 0.01$.

Data information: GFP = eGFP; ONL, outer nuclear layer; INL, inner nuclear layer. (F) Detailed statistical analysis in Appendix Table S1.
Source data are available online for this figure.

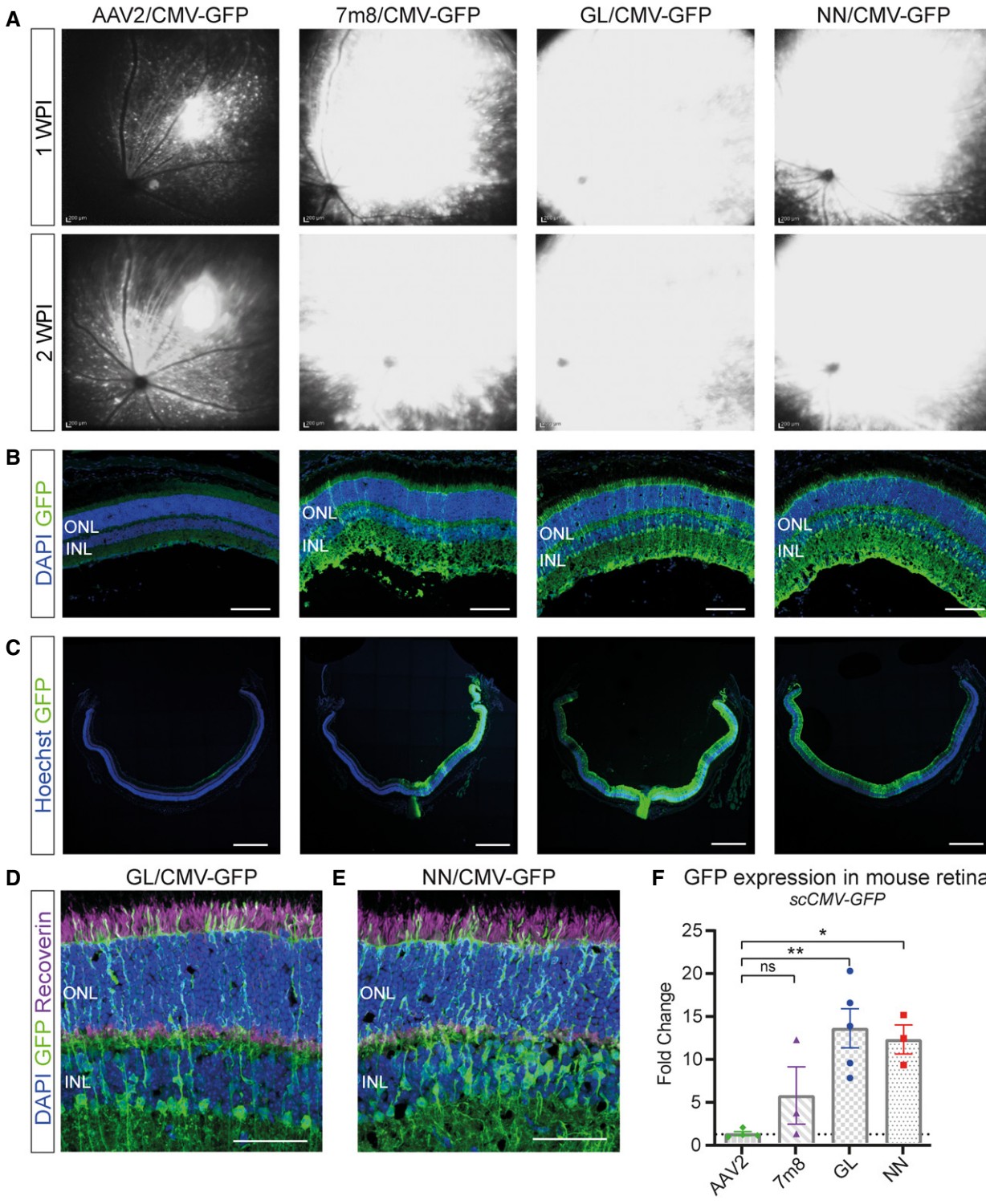

**Figure 1.**

heparin affinity chromatography with gradient elution was performed. As suggested by the sequence, indeed both AAV2.GL and AAV2.NN bind to heparin, however with lower affinity than AAV2 as they elute at a lower salt concentration, i.e. earlier from the column (Fig EV3C). In addition to virion capturing by surface

proteoglycans, inefficient uncoating, i.e. the release of vector genomes into the nucleus has been proposed as a transduction barrier (Horowitz et al, 2013; Rossi et al, 2019). As an indirect measure of uncoating, a thermal stability assay was performed (Horowitz et al, 2013; Rossi et al, 2019) showing that most AAV2

particles are still intact at 63°C (step 7), while AAV2.GL and AAV2.NN particles were already disassembled at this temperature (Fig EV3D) suggesting their genome is released under less harsh conditions. Finally, we compared the novel capsids to AAV2 with regard to nuclear transport efficiency. Both localise within the nucleus with twofold (AAV2.NN) and fourfold (AAV2.GL) higher efficiency compared with AAV2 (Fig EV3E). As the peptide insertions not only mediate cell transduction but also impact the structure of the second highest peak (Fig EV1A), thereby affecting recognition by pre-existing neutralising antibodies, we performed a neutralisation assay with human serum (Fig EV3F). Both novel

capsids, particularly AAV2.GL, were less sensitive to neutralisation compared with AAV2, a promising feature in light of the neutralising antibody prevalence in the human population. To obtain some insight into the possible structural changes induced by the peptide insertions, we performed comparative modelling of the novel capsids based on the AAV2 structure (PDB 6IH9) (Fig EV4A and B) using Robetta (Raman *et al*, 2009; Song *et al*, 2013). The resulting models were aligned to the reported structures of parental AAV2 (PDB 6IH9) (Fig EV4C–F) and of AAV2.7m8 (PDB 6U0R) (Fig EV4G and H). This *in silico* modelling revealed striking similarities between AAV2.GL and AAV2.NN, as well as clear differences of

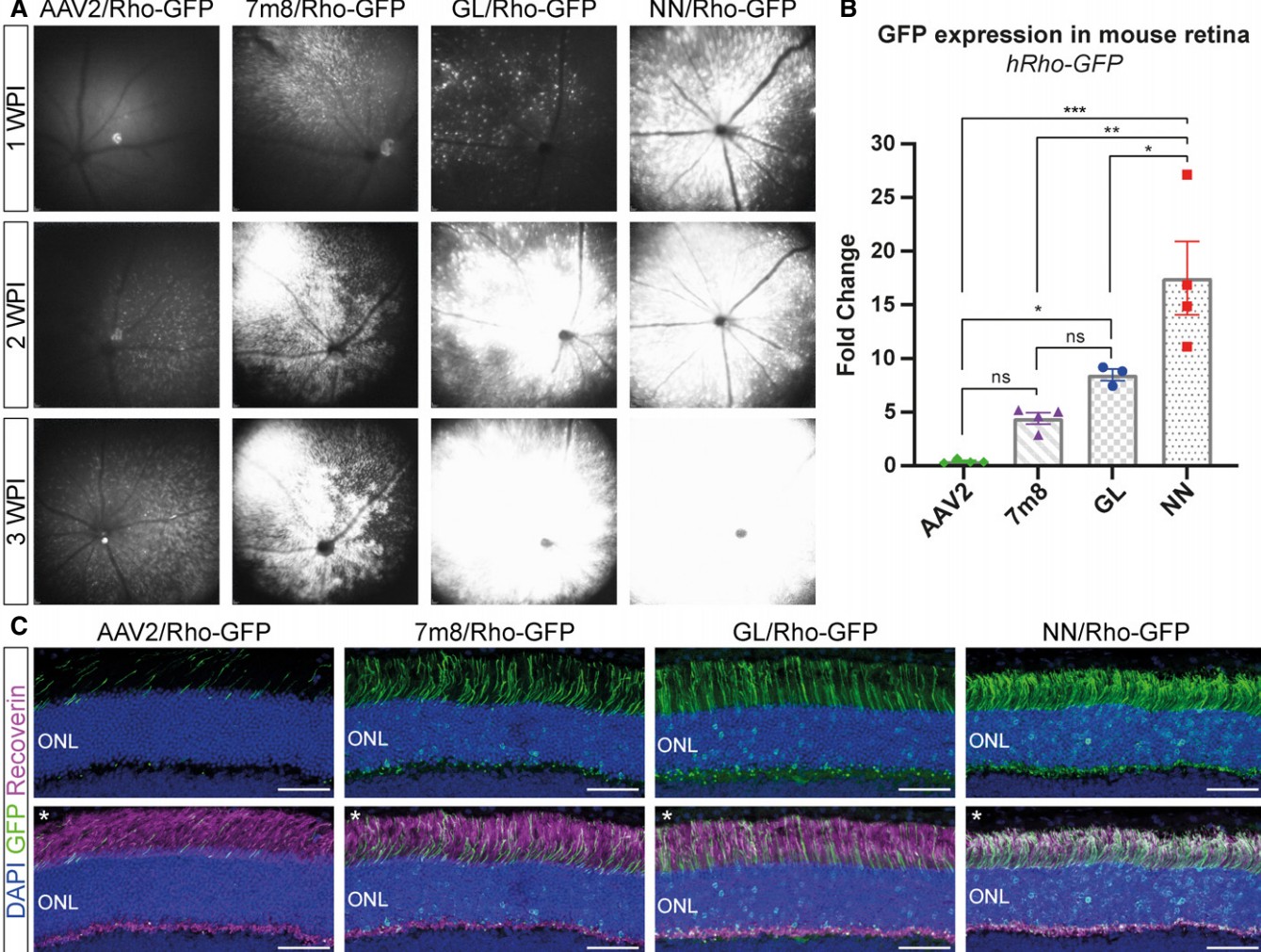

**Figure 2. Novel capsids outperform AAV2 and AAV2.7m8 in mice using photoreceptor-specific promoter.**

Vector mediated eGFP expression after intravitreal injection of novel capsid variants compared with controls in 2-month-old mice.

A *In vivo* cSLO examinations of the mouse retina at 1-, 2- and 3-week post-injection (WPI) of 8E9 total vg of ss-hRho-eGFP packaged with parental serotype AAV2, state-of-art AAV2.7m8, novel AAV2.GL and AAV2.NN capsid variants.

B qRT–PCR data comparing fold change in eGFP transcript levels relative to ITR2 amplicons from retinal samples infected with all four capsid variants. Biological replicates $n \geq 3$. Bars show mean ± SEM. One-way ANOVA was performed with post hoc Holm–Sidak multiple comparisons test. *$P \leq 0.05$, **$P \leq 0.01$, ***$P \leq 0.001$.

C Confocal scans of retinal cross sections, with focus on the ONL, immunolabelled for DAPI, eGFP and recoverin. The upper panel shows only the DAPI and eGFP signal in the ONL, and the lower panel shows the same images (noted with asterisk *) with merged DAPI, eGFP and recoverin signal. Acquisition settings were kept constant for all samples. Scale bar: 50 μm.

Data information: GFP = eGFP; ONL, outer nuclear layer. (B) Detailed statistical analysis in Appendix Table S1.
Source data are available online for this figure.

both novel capsids when compared to AAV2 or AAV2.7m8. Specifically, this modelling data suggest that the loop extension caused by the peptide insertion in AAV2.GL and AAV2.NN is oriented differently than in AAV2.7m8.

### Novel vectors achieve widespread transduction in canine retina

To test whether the novel variants are functional in species which better approximate human eye size and physiology, we assessed their transduction profile after intravitreal injection in dog eyes. Five 10-month-old beagles received a single intravitreal injection containing 2E11 total vg of sc-CMV-eGFP packaged with AAV2, AAV2.GL or AAV2.NN capsids and monitored over a 6-week period. As in the mouse study, the treated dogs were followed up using cSLO fundus fluorescence imaging. Intraocular inflammation developed at 2 and 3 weeks post-injection in two animals. The inflammation was independent of the vector capsid used, as the affected eyes had been treated with AAV2, AAV2.GL or AAV2.NN. The two affected dogs were euthanized at 4 weeks post-injection; the remaining three animals showed no evidence of inflammation and were euthanized at 6-week post-injection. AAV2 resulted only in weak and sparse eGFP fluorescence primarily across the nerve fibres, while both novel capsids produced very strong and broadly distributed eGFP signal in the dog retina after 4, 5 and 6 WPI (Fig 3A). This signal was stronger in the central/nasal regions of the fundus compared with temporal, as seen previously (Boyd *et al*, 2016b). Confocal scans of retinal cross sections revealed widespread eGFP signal (Fig 3B) and penetrance throughout all retinal layers, as well as retinal-pigmented epithelium (RPE), for both AAV2.GL and AAV2.NN (Fig 3C). In the ONL, eGFP signal co-localised with rhodopsin and M-opsin expression for both novel capsid variants (Fig 3D and E), confirming photoreceptor transduction. In contrast, dog eyes injected with AAV2 had only a few eGFP-positive cells in the retinal ONL (Fig 3C).

### Novel vectors transduce non-human primate fovea and retinal periphery

The evaluation of AAV2.GL and AAV2.NN capsids was further expanded using NHP eyes since their retinal anatomy and physiology are the closest to human. The first of the two animals was intravitreally injected with 1E12 total vg of sc-CMV-eGFP packaged with AAV2.NN capsid to the right eye and 1E12 total vg of the same transgene packaged with AAV2.GL capsid to the left eye. To avoid axial bias and assess low-volume delivery, the second animal received in both eyes 39 μl 1E12 total vg of AAV2.GL intravitreally.

The animals were monitored weekly via cSLO throughout the study to visualise transduction efficiency in the form of eGFP expression. *In vivo* cSLO imaging at 8 WPI revealed for both novel capsids a strong eGFP signal in the fovea, around major blood vessels throughout the retina and towards the periphery (Fig 4A and B). After euthanasia at 8 WPI, the NHP eyes were extracted and prepared for histological analyses of the retinae. IHC staining and confocal microscopy for eGFP revealed an excellent penetrance profile of both novel capsid variants into all retinal layers, particularly at the fovea centralis (Fig 4C and D) where a mean 54.5% and 54.3% of photoreceptors were eGFP$^+$ in the AAV2.GL- and AAV2.NN-treated retina, respectively (AAV2.GL technical $n = 4$, AAV2.NN technical $n = 3$) (Fig 4E). The punctate signal recorded via cSLO was observed also in the peripheral NHP retina slices (Fig 4F and G) with multiple loci showing eGFP$^+$ photoreceptors, which are seen more clearly at higher magnification (Fig 4H and I). This showed efficient transduction of NHP retinas via intravitreal delivery using novel capsids AAV2.GL and AAV2.NN.

### Novel vectors transduce human photoreceptors

Next, we aimed to evaluate the transduction efficacy of AAV2.GL and AAV2.NN in the human retina. To this end, 1E11 vg of the capsid variants packaged with sc-CMV-eGFP were locally applied to the vitreal side (inner limiting membrane side) of freshly isolated and cultivated human retinal explants ($n = 3$) (Fig 5A). Human retinal explants were kept in culture for 10 days after transduction and eGFP expression was tracked during this time period via epifluorescence microscopy. The *en face* view epifluorescence images at days 3, 6 and 9 post-infection revealed a broad eGFP signal across the entire retinal explant, which gradually increased over time for both capsid variants (Fig 5B), indicating the increasing expression levels of the transgene. IHC analysis at 10 days post-infection revealed robust expression of eGFP, which was most prominent in the photoreceptors of the human retinas for both AAV2.GL and AAV2.NN (Fig 5C).

### Intravitreal gene supplementation using AAV2.GL restores vision in Cnga3$^{-/-}$ mice

To evaluate the efficiency of the newly engineered capsids in a relevant preclinical disease model, we performed a *proof-of-concept* gene supplementation study in the *Cnga3*$^{-/-}$ mouse model of achromatopsia (Biel *et al*, 1999) using the cone-favouring AAV2.GL capsid (Table 1). 2-week-old *Cnga3*$^{-/-}$ mice received a single intravitreal injection of 1E10 vg (1 μl) AAV2.GL carrying the murine

---

**Figure 3. AAV2.GL and AAV2.NN transduce the dog retina more efficiently than AAV2.**

Vector mediated eGFP expression after intravitreal injection in 10-month-old dog eyes.

A    *In vivo* cSLO examinations of the dog retina at 4, 5 and 6 weeks post-intravitreal injection of 2E11 total vg (in 200 μl) of sc-CMV-eGFP packaged with wild-type AAV2 (right eye), AAV2.GL (right eye) and AAV2.NN (left eye) capsid variants.

B, C    Confocal scans of immunolabelled retinal cross sections (14 μm) stained for GFP and DAPI at 6 WPI time point.

D, E    Confocal scans of immunolabelled retinal cross sections (14 μm) at 6 WPI time point, focusing on the photoreceptors, using antibodies against eGFP, Rhodopsin (Rho) and M-opsin (MO).

Data information: DAPI was used a nuclear marker. Scale bar panel (B): 200 μm, (C-E): 50 μm. Acquisition settings were kept constant for all samples. GFP = eGFP; RPE, retinal-pigmented epithelium; OS, outer segments; ONL, outer nuclear layer; INL, inner nuclear layer; GCL, ganglion cell layer.

Source data are available online for this figure.

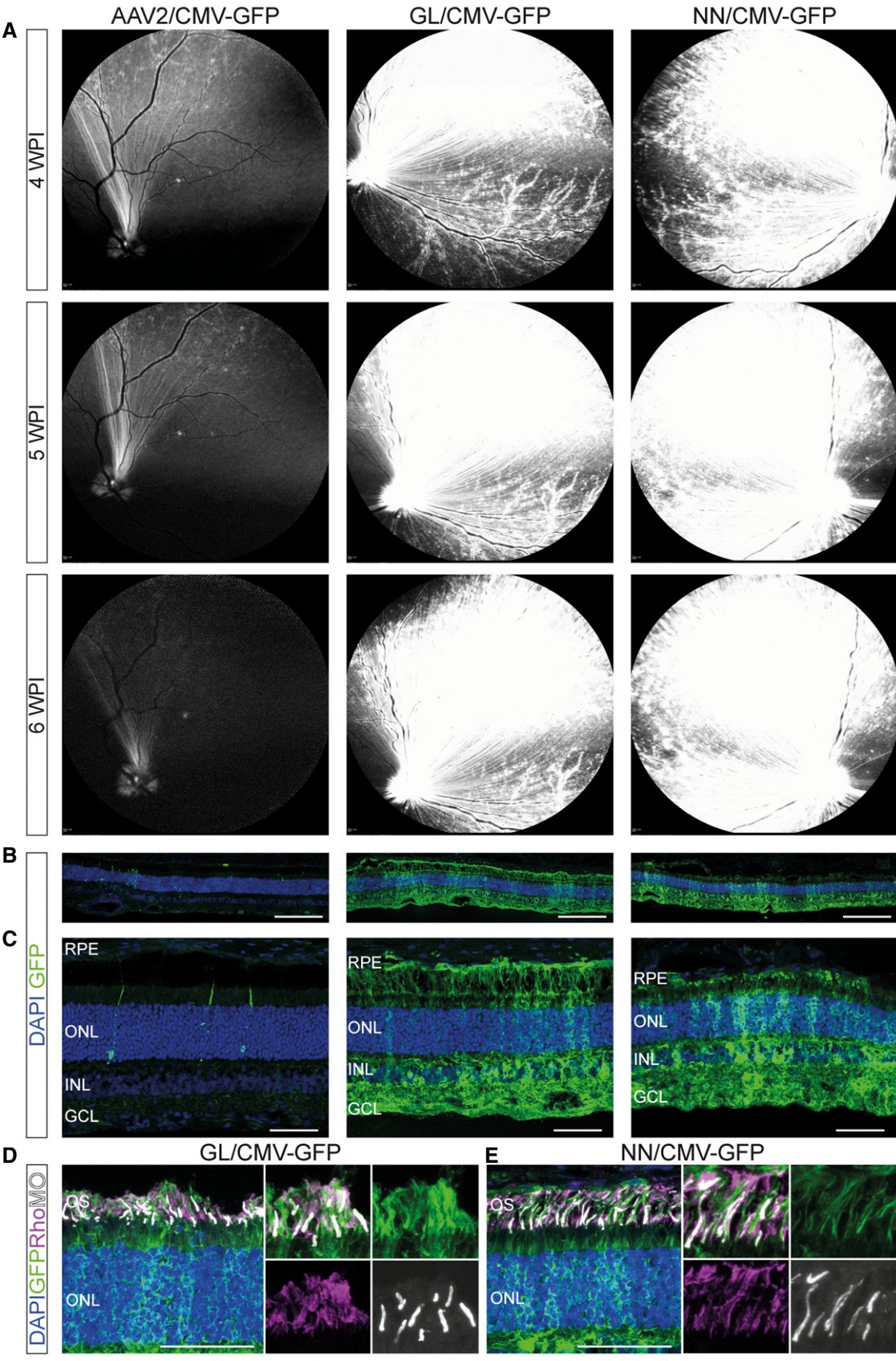

**Figure 3.**

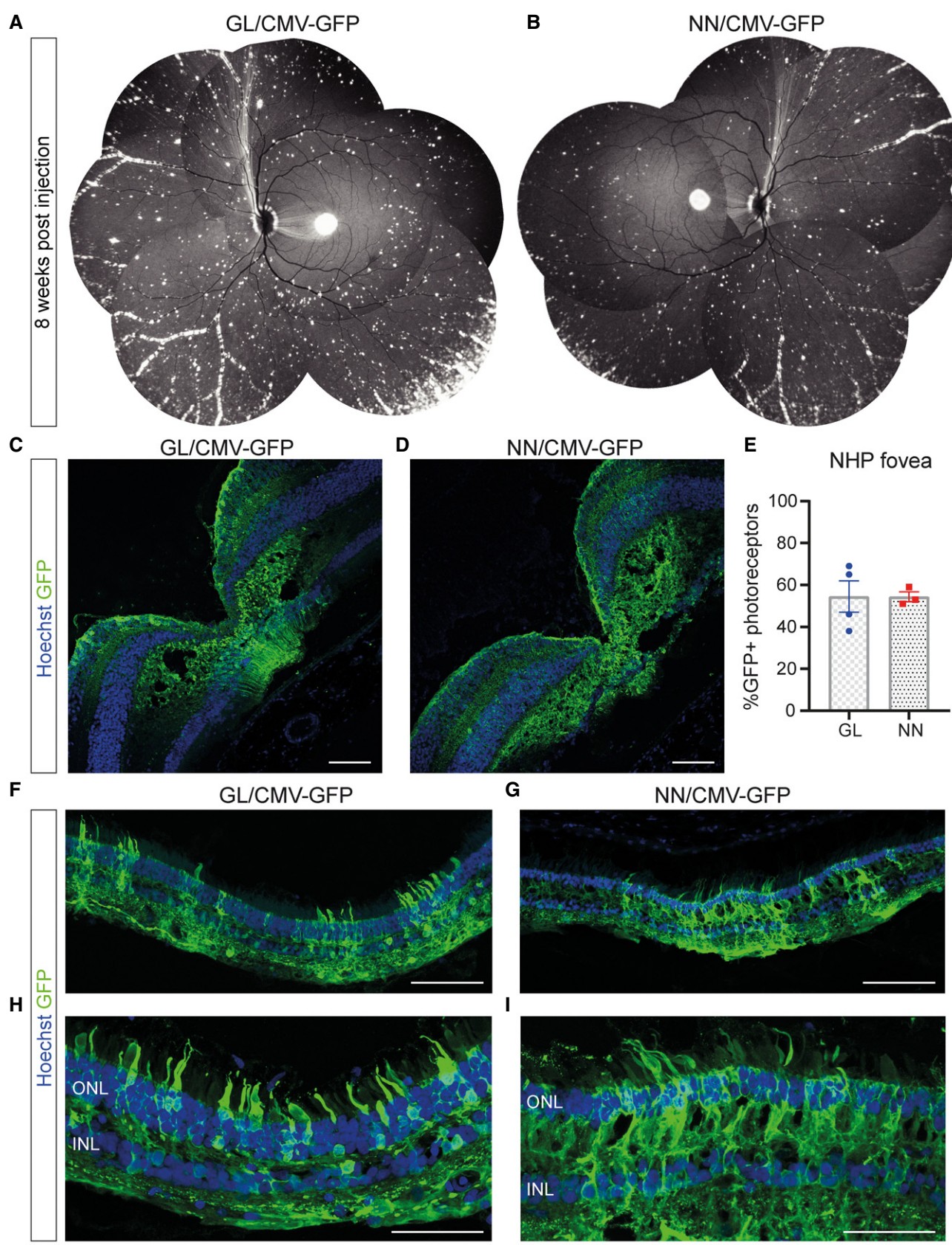

**Figure 4.**

**Figure 4. AAV2.GL and AAV2.NN efficiently transduce the non-human primate retina.**

Vector mediated expression after intravitreal injection in ~3-year-old cynomolgus macaque eyes.

A, B    In vivo cSLO examination images collaged to provide NHP fundus overview at 8 WPI of E12 total vg (in 38.9–100 µl) of sc-CMV-eGFP packaged with AAV2.GL and AAV2.NN capsid variants, respectively.

C, D    Confocal scans of immunolabelled retinal cross sections (14 µm) of the fovea centralis from AAV2.GL- and AAV2.NN-treated animals, respectively.

E    Percentage quantification of eGFP$^+$ photoreceptors in the ONL of foveal cross sections. AAV2.GL technical repeats $n = 4$, AAV2.NN technical repeats $n = 3$. Bars show mean ± SEM.

F, G    Confocal scans of immunolabelled retinal cross sections (14 µm) of the periphery from AAV2.GL (F)- and AAV2.NN (G)-treated retinae.

H, I    Higher magnification confocal scans of immunolabelled peripheral retina.

Data information: Scale bar for (C-D), (F-G): 100 µm, (H-I): 50 µm. Acquisition settings were kept constant for all samples. ONL, outer nuclear layer; INL, inner nuclear layer; GCL, ganglion cell layer.

Source data are available online for this figure.

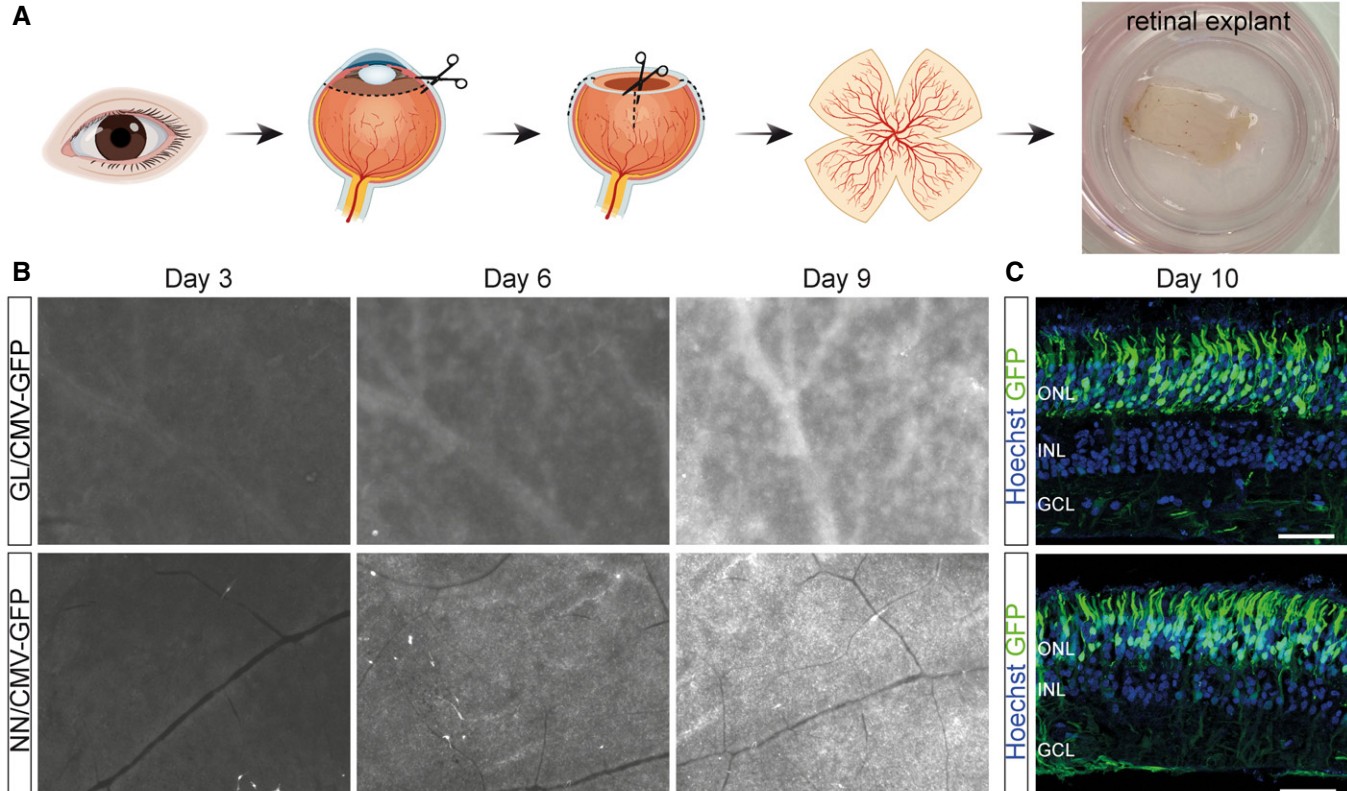

**Figure 5. AAV2.GL and AAV2.NN efficiently transduce human photoreceptors.**

Vector mediated eGFP expression in human retinal explants.

A    Schematic of retinal explant culture procedure.

B    EVOS epifluorescence images at 2× magnification of the human retinal explants at 3, 6 and 9 days *in vitro* (DIV) after transduction with 1E11 total vg of sc-CMV-eGFP packaged with AAV2.GL and AAV2.NN capsid variants. White signal indicates eGFP.

C    Confocal scans of day 10 tissue sections immunolabelled for eGFP and Hoechst. Scale bar: 50 µm.

Data information: Acquisition settings were kept constant for all samples. ONL, outer nuclear layer; INL, inner nuclear layer; GCL, ganglion cell layer.

Source data are available online for this figure.

*Cnga3* gene under the control of a murine short-wavelength opsin promoter (mSWS) (Michalakis *et al*, 2010). Restoration of cone-mediated light responses was investigated using electroretinography (ERG). The light responses of the retina were assessed at 10 WPI via photopic ERG, showing restored cone-mediated flicker (Fig 6A) and single-flash responses (Fig 6B) compared with age-matched untreated

*Cnga3*$^{-/-}$ animals. The B-wave amplitudes in eyes treated with AAV2.GL/mSWS-mCnga3 were significantly higher for all three illuminances, namely 1 cd.s/m$^2$ ($P = 0.005284$), 3 cd.s/m$^2$ ($P = 0.000029$) and 10 cd.s/m$^2$ ($P = 0.000006$), although these were still ~ 1 order of magnitude lower than wild-type responses (Fig 6C). After euthanasia, the presence of Cnga3 was confirmed via IHC, where sections

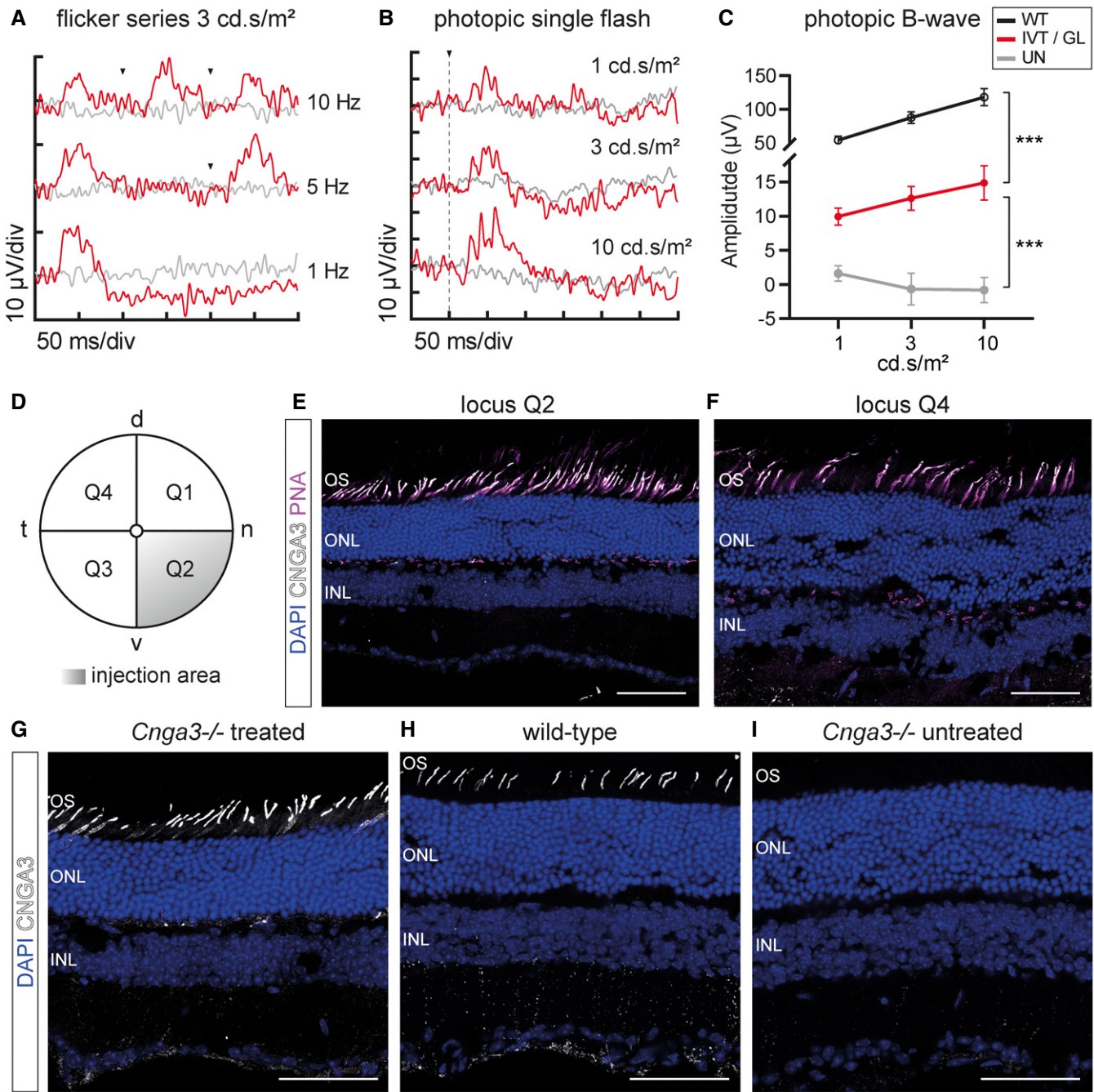

**Figure 6. Intravitreal gene supplementation with AAV2.GL restores cone function in *Cnga3*$^{-/-}$ mice.**

Restoration of cone function following intravitreal delivery of 1E10 vg AAV2.GL/mSWS-mCNGA3 therapeutic vector.

A  Representative photopic ERG flicker series at 1, 5 and 10 Hz with 3 cd.s/m$^2$ illumination at 10 WPI. Arrowheads indicate light stimulus.

B  Representative photopic single-flash ERG signal with 1, 3 and 10 cd.s/m$^2$ illumination. The vertical dotted line indicates light stimulus.

C  Quantification of photopic B-wave amplitude from single-flash responses to 1, 3 and 10 cd.s/m$^2$ illumination of wild-type *C57BL6/J x 129/sv* eyes (WT, n = 3) compared with treated eyes (IVT/GL, n ≥ 3) and untreated eyes (UN, n ≥ 3) at 10 weeks post-injection. Symbols show mean ± SEM. Ordinary two-way ANOVA with α = 0.05 revealed significant difference (***$P < 0.0001$) between samples. Additional analysis with multiple t-tests using the Holm–Sidak method, α = 0.05, assuming sample populations have the same scatter also revealed significant difference between samples at individual illuminations.

D  Graphical representation of mouse fundus (d = dorsal, n = nasal, v = ventral, t = temporal) divided into quadrants. Q2 indicates locus of intravitreal injection.

E, F  Confocal images of sections proximal to the injection site in Q2 (E) and distal to the injection site in Q4 (F), showing Cnga3 expression in PNA-positive cone outer segments.

G–I  Confocal images of *Cnga3*$^{-/-}$-treated (G), wild-type (H) and *Cnga3*$^{-/-}$-untreated (I) retinal sections stained for Cnga3. Note some non-specific signal detected in INL and adjacent synaptic layer. Scale bar: 50 μm.

Data information: OS, outer segment layer; ONL, outer nuclear layer; INL, inner nuclear layer. (C) Detailed statistical analysis in Appendix Table S1.
Source data are available online for this figure.

proximally and distally to the injection site (Fig 6D) were stained for Cnga3 and the cone photoreceptor marker peanut agglutinin (PNA) (Fig 6E and F). Confocal microscopy confirmed widespread transduction and successful wildtype-like expression of Cnga3 protein in PNA-positive cone photoreceptor outer segments throughout the retina. To further validate the protein expression profile, we compared *Cnga3$^{-/-}$*-treated retina (Fig 6G) to wild-type (Fig 6H) and *Cnga3$^{-/-}$*-untreated retina (Fig 6I), thus confirming the correct CNGA3 localisation and distribution. This highlights the potential of AAV2.GL as a vector for effective gene delivery using an administration route with less collateral damage.

## Discussion

Multiple gene therapies for monogenic ocular diseases are rapidly advancing to clinical trials with the first approved drug, voretigene neparvovec, already being administered in the clinic (Darrow, 2019). For a thorough review on clinical trials in retinal dystrophies, see (Trapani & Auricchio, 2018). While an important medical advancement, rAAV vectors currently used in the clinic use capsids of natural serotypes which pose several limitations on efficiency, as seen in clinical trials for Leber's congenital amaurosis (LCA2) patients (Bainbridge *et al*, 2008; Hauswirth *et al*, 2008; Maguire *et al*, 2008) where photoreceptor degeneration continued even after subretinal gene therapy with an rAAV2-RPE65 vector (Cideciyan *et al*, 2013; Bainbridge *et al*, 2015; Jacobson *et al*, 2015). This shortcoming could be apportioned to the local administration method, which can be deleterious and often only achieves local transduction and, therefore, therapeutic effect. While the option of multiple subretinal injections remains, the risks associated with detaching an already compromised retina at multiple sites overweighs the benefits. Furthermore, the expertise and equipment required for subretinal injections, currently only performed by highly trained vitreoretinal surgeons at specialised centres, may limit the treatment availability. Notably, the accuracy of subretinal injections has been challenged in terms of actual volume delivered versus intended volume delivered (Hsu *et al*, 2018). Alternative injection methods have been attempted, such as sub-inner limiting membrane injection (Gamlin *et al*, 2019) or suprachoroidal injection (Ding *et al*, 2019; Yiu *et al*, 2020), which could overcome risks associated with retinal detachment. Nevertheless, intravitreal delivery of retinal gene therapies could be offered with potentially greater precision by a larger pool of professionals and without the need for specialised equipment. The development of novel rAAV vectors, capable of deep penetration of the retina layers after intravitreal administration, as we report here, could considerably improve patient outcome.

While we continue to improve our understanding of AAV trafficking and how this is facilitated by capsid-host cell interactions (Summerford *et al*, 2016; Dudek *et al*, 2018; Hamilton *et al*, 2019; Dudek *et al*, 2020), bottom-up capsid engineering relies on selective pressures applied to capsid libraries, in order to generate superior vectors for targeting specific cell populations. In this study, a library of peptide display-diversified AAVs based on AAV2 was screened in *C57BL6/J* mice. The library was subjected to a genotype–phenotype coupling step, which enabled the generation of uniform capsids, displaying the 7-mer on the capsid

surface by each of the 60 subunits (Figs EV1B and EV4A and B) for differential cellular contact. Capsid variants with strong or wild-type AAV2-like affinity for HSPG were excluded before *in vivo* screening by counter-selecting the library on a heparin affinity column. HSPG serves as natural primary receptor of AAV2 (Kern *et al*, 2003; Opie *et al*, 2003) and is responsible for broad tropism and limited spreading of AAV2. Intriguingly, it appears that the ability to bind HSPG, albeit with a lower affinity than AAV2, is critical for retinal transduction through the vitreous and therefore fostered the selection of variants with respective features from our library, agreeing with previous reports (Woodard *et al*, 2016). Structural modelling of the novel capsids (Fig EV4A and B) using Robetta (Raman *et al*, 2009; Song *et al*, 2013) and comparison with the parental AAV2 shows a shift in the geometry of the hypervariable loop 4 on the capsid surface (Fig EV4C and D). This loop forms the second highest peak at the threefold symmetry axis of the AAV2 capsid and is part of the HSPG-binding motif as well as a target of neutralising antibodies. Similar to our novel capsids, the insertion of LALGETTRP peptide in AAV2.7m8 extended this loop and changed its structure (Bennett *et al*, 2020). Based on our *in silico* modelling, the loop extensions of both novel variants appear to be oriented in the opposite direction compared with AAV2.7m8. This however is subject to confirmation via structural analysis, as recent work indicated that such peptide-extended loops can be flexible in 3D space (Bennett *et al*, 2020).

Canonical HSPG binding to R585 and R588 (Kern *et al*, 2003) is disturbed by the peptide insertion because it separates the two residues. However, the arginine residue at position 7 of the inserted peptide in AAV2.GL and AAV2.NN forms in conjunction with the two alanine residues of the linker a new binding motif together with R588 of the parental AAV2 sequence (Perabo *et al*, 2006b; Uhrig *et al*, 2012). This motif is accessible for receptor binding according to the modelled structure (Fig EV4E and F). Furthermore, the two alanine residues (instead of GN) or the difference in the loop structure might cause the observed lower affinity to heparin compared with AAV2. Another possibility is that the deamidation status of N587 has changed as a result of the peptide secondary structure (Fig EV4E and F) blocking accessibility to the residue. Deamidation of asparagine residues has been associated with reduced transduction of AAV8, which was partially reversed when the $+1$ position was switched to alanine (Giles *et al*, 2018). Our novel capsids have three alanine residues proceeding N587. Although the backbone was AAV2, asparagine deamidation has been reported for various serotypes (Giles *et al*, 2018), indicating that this modification is independent of serotype and could play a role in the improved potency of AAV2.GL and AAV2.NN. Irrespective of the induced modifications, there was no impact on vector yield as AAV2.GL and AAV2.NN were produced at titres similar or higher than AAV2 and AAV2.7m8 (Fig EV3A and B); a promising feature for clinical translation. By comparing our *in silico* models to the AAV2.7m8 structure (Bennett *et al*, 2020) (Fig EV4G and H), it becomes evident that a capsid diversification strategy with distinct starting libraries and screening approaches can result in structurally different capsid variants, though with similar capabilities. This means that library-borne engineered vectors will ultimately reflect the screening criteria used to distil them, giving us powerful new tools, even though

only a fraction of the theoretical variants in a library were experimentally screened. This phenomenon is starting to be addressed by machine-guided AAV design (Ogden et al, 2019).

Our library was administered intravenously, and thus, variants had to pass multiple extracellular barriers to reach the retina within only 24 h. Specifically, virions had to escape systemic clearance and to penetrate through the blood-vessel endothelial and blood–retina barriers (BRB). Importantly, also intracellular barriers had to be crossed for effective gene expression in photoreceptors. Upon cell entry, the vector particles must traffic through and escape the endolysosomal pathway, shuttle through nuclear pores and uncoat their genome to enable transgene expression (Ding et al, 2005). The novel variants were able to cross these intracellular barriers with greater efficiency than AAV2, as shown by the higher levels of nuclear localisation (Fig EV3E) and higher efficiency in transgene expression. As the intention was to engineer superior rAAV vectors for ocular application, the top five candidates found in cones or rods (Table 1) were validated for retinal penetrance after intravitreal injection in mice (Fig EV2), revealing AAV2.GL and AAV2.NN as superior. These were vectorised and shown to outperform the parental AAV2 after intravitreal delivery, with regard to both qualitative (Figs 1A–C and 2A) and quantitative reporter expression (Figs 1F and 2B). Furthermore, we demonstrated the potency of the novel capsids when administered subretinally instead of intravitreally (Fig EV5). Both novel capsids resulted in high eGFP expression outside the subretinal bleb area, indicating that similar to previously reported engineered rAAVs (Khabou et al, 2018; Boye et al, 2020), our AAV2.GL and AAV2.NN support lateral spreading. Nevertheless, as vector lateral spreading cannot compensate for all other risk factors associated with subretinal application, we focused on validating these two novel capsids for their intravitreal penetrance in larger mammals and human tissue as they are more clinically relevant models.

A pertinent issue with screening for novel AAV capsids in C57BL6/J mice is whether the distilled capsids are exclusively potent in mice, or whether their efficiency translates across species (Hordeaux et al, 2018). Indeed, there is often interspecies variation in the potency of rAAV vectors in the retina, potentially as a result of distinct morphological features (Dalkara et al, 2009) as well as the presence of neutralising antibodies in dogs (Boyd et al, 2016a) and in primates (Kotterman et al, 2015). We show that both AAV2.GL and AAV2.NN had a similarly improved in vivo efficiency in dogs (Fig 3) and non-human primates (Fig 4). In dogs, reporter expression was coinciding but not restricted to blood-vessel colocalisation (Fig 3B and C); a pattern often observed in larger animals as the ILM is less compact in those loci. This finding was partially replicated in non-human primates as a strong punctate signal was detected in the periphery of the fundus (Fig 4A and B). However, stronger eGFP signal was observed close to major blood vessels in this species. The macula and fovea centralis were also strongly transduced (Fig 4C and D), which is apportioned to the specialised retinal lamination at the fovea, and speaks for the clinical applicability of the novel capsids to target cone-specific dystrophies (Fig 4E). The final model used to test AAV2.GL and AAV2.NN was human retinal tissue kept ex vivo in the form of retinal explant cultures, which is arguably the most informative of all models in terms of transduction of human cells (Murali et al, 2019). Indeed, retinal explants are missing relevant in vivo barriers such as the ILM and

therefore cannot exclude that viral transduction in the human retina could differ. Nevertheless, we demonstrate that AAV2.GL and AAV2.NN can transduce human retinal cells, with a particularly high efficacy in human photoreceptors (Fig 5C).

In addition to eGFP reporter-based studies, we assessed the efficacy of AAV2.GL in a preclinical proof-of-concept study for intravitreal gene supplementation in degenerating cones of the $Cnga3^{-/-}$ mouse model of achromatopsia. While we cannot exclude variable efficiencies in different mouse strains, we provide evidence for the C57BL6/J and mixed background, as the $Cnga3^{-/-}$ mice in our study were C57BL6/J x 129/sv mixed. By choosing a mouse model of cone photoreceptor degeneration, the challenge of detecting any rescue effect was greater as cones are already limited in number in the mouse retina (~3% of all photoreceptors) (Peirson et al, 2018). As such, successful visual restoration would suggest it was possible to target and revive a very scarce cell type. Gene supplementation via the less invasive intravitreal route rescued cone photoreceptor-driven photopic ERG responses compared with untreated eyes (Fig 6A–C) and resulted in widespread expression of Cnga3 protein in cone photoreceptors (Fig 6E and F). Although ubiquitous promoters are known to outperform cell-specific promoters (McClements & MacLaren, 2013), we show that our novel capsids effectively transduce cones and rods after intravitreal administration using photoreceptor-specific promoters, such as mSWS (Fig 6) and hRho (Fig 2), respectively. Efficient gene delivery via the intravitreal route has significant clinical implications, as clinical trials could expand patient recruitment to non-specialised centres, making treatment more accessible. While this holds promise for future clinical application, similar experiments in larger animal models and with additional cell type-specific promoters are needed to explore the full potential of the novel vectors. In addition to applications for gene therapy of IRDs, the widespread retinal transduction properties of AAV2.GL and AAV2.NN render them optimal vectors for therapies against more common or acquired retinal disorders such as age-related macular degeneration, diabetic macular oedema or diabetic retinopathy.

For clinical translation, a careful consideration of immunogenicity is necessary. To this end, effects of various vector-specific factors such as the exact vector genome composition (e.g. CpG content), the immunogenic potential of the encoded transgene or production-related factors need to be considered. It is positive that although our study was not designed to investigate the immune responses of AAV2.GL and AAV2.NN, they produced no indication of an increased immunogenic potential. In contrast, both variants appear to be less sensitive to pre-existing antibodies (Fig EV3F). The inflammation events we observed in 2 dogs in our study were independent of the vector capsid used for injection and we hypothesise that this was in response to eGFP. In a previous study of different AAV2 capsid variants delivered in the dog by intravitreal injection, inflammation was also detected in some dogs and a study of peripheral blood mononuclear cells showed a robust response to eGFP but not to the capsid (Boyd et al, 2016b). Immune response to the widely used reporter gene eGFP is a recognized issue (Stripecke et al, 1999). As such, further studies dedicated to immune responses are needed to fully characterise the novel vectors. Should it be necessary, AAV2.GL and AAV2.NN could be combined with other technologies, such as rational design-based modification of immunogenic residues at the capsid surface or optimisation of the

AAV2 genome in order to mitigate any risk of immune responses (Perez *et al*, 2020).

So far, it is unclear which receptors mediate the specific tropism of our novel variants. While the unique 7-mer peptides of AAV2.GL and AAV2.NN differ in amino acid positions, both sequences end with a P-S/T-R motif. This sequence on its own or in combination with the linker sequence (AA) and additional residues of the 7-mer and/or the subsequent AAV2-VP1 sequence (RQ) could participate in specific receptor interactions. While it is beyond the scope of our here reported study, it is in principle possible to identify receptors targeted by AAV peptide display-selected capsid variants (Sallach *et al*, 2014).

Taken together, this work introduces two novel potent AAV capsid variants, which expand our current toolbox for gene delivery in research and clinical application. Making intravitreal gene delivery a common practice, by using vectors such as AAV2.GL and AAV2.NN, can revolutionise the therapeutic impact and accessibility to gene therapy for patients with inherited or acquired retinal dystrophies.

# Materials and Methods

### Study approval

All animal experiments were performed according to the ARVO statement for the use of animals in ophthalmic and vision research and were approved by the local authorities (Regierung von Oberbayern, Michigan State University and Charles River Laboratories Institutional Animal Care and Use Committees). The use of human tissue samples was approved by the institutional review board of the Ludwig-Maximilians-University Munich (Project number 17-531) and conformed to the principles set out in the WMA Declaration of Helsinki and the Department of Health and Human Services Belmont Report. Written informed consent was obtained from patients before the tissue samples were collected and used for investigational purposes.

### Animals

Male and female adult (2- to 3-month-old) *C57BL6/J* or *C57BL6/J x 129/sv* wild-type mice, *RG-eGFP* mice expressing eGFP under control of the cone-specific RG promoter (Fei & Hughes, 2001) and 2-week-old *Cnga3*$^{-/-}$ mice (Biel *et al*, 1999) were used in this study. *RG-eGFP* mice were kindly provided by Drs. Y. Fei and T.E. Hughes (Yale University School of Medicine). Five male and female 10-month-old beagles and two approximately 3-year-old male cynomolgus macaques were used for the large animal studies. Mice, dogs and NHP were housed under standard white cyclic lighting conditions and were used irrespective of gender.

### AAV libraries and vector production

24 h after intravenous injections of rAAV libraries, mice were euthanized by cervical dislocation and their retinas were harvested. Nuclei were isolated from whole retina using the Subcellular Protein Fractionation Kit for Tissues (87790, Thermo Fischer). DNA was isolated using the DNeasy Blood & Tissue Kit (69504, Qiagen) and

assayed by qPCR using *cap*-specific primers (Rossi *et al*, 2019). In addition, total DNA was isolated from retina using the DNeasy Blood & Tissue Kit (69504, Qiagen), amplified by PCR with primers 5'-GTATCTACCAACCTCCAGAGAG-3' and 5'-GTGTTGACATCTG CGGTAGC-3' and cloned into the plasmids pWt.oen or pLG to generate a plasmid pool for sublibrary production. NGS analysis was performed on 454-pyrosequencing platform (GS Junior, Roche). For the generation of helper plasmids for rAAV vector production, sense and antisense oligonucleotides with corresponding peptide insert sequences including linkers were synthesized (Eurofins Genomics GmbH). Oligos were hybridized to form double-stranded DNA oligos and then cloned into pRC99 (Girod *et al*, 1999; Nicklin *et al*, 2001) encoding AAV2 *rep* and *cap* genes resulting in the generation of AAV2 *cap* open reading frame with the intended peptide insertions at I-587 of VP1. The insertion of the sequence destroys the MluI / AscI sites and generates a novel EagI site which can be used for screening of correct clones. Vector production was performed as previously described (Becirovic *et al*, 2016). A self-complementary (sc) rAAV plasmid containing a CMV-eGFP expression cassette (Hacker *et al*, 2005) and single-strand rAAV plasmids containing a hRho-eGFP or mSWS-mCnga3 expression cassette (Michalakis *et al*, 2010) were used as *cis* plasmids.

### Retinal harvesting and photoreceptor isolation

For retinal isolation from euthanized mice, each eye was protruded by placing Dumont #7 curved forceps (Fine Science Tools) around the rear part close to the optic nerve and applying gentle pressure. The cornea was cut along the equator with a sharp blade. Subsequently, the retina was detached from the retinal pigment epithelium (RPE) and removed from remaining eye tissue, together with the lens and the vitreous body, by gently pulling the forceps upwards. After rinsing the tissue in ice cold 0.1M phosphate buffer (PB) and removing the lens and any remaining RPE cells with the forceps, the two retinas per mouse were pooled and transferred into an Eppendorf tube, snap frozen in liquid nitrogen, and stored at −80°C. For isolation of rod or cone photoreceptors, retinas were dissociated as previously described (Feodorova *et al*, 2015). Cone photoreceptors positive for eGFP were isolated from *RG-eGFP* retinas by fluorescence-activated cell sorting (FACS) using a FACSAria II system (BD Biosciences). Rod photoreceptors were enriched by magnetic-activated cell sorting (MACS) using the Dynabeads Protein G Immunoprecipitation Kit (10007D, Thermo Fisher) in combination with a rat anti-mouse CD73 antibody (1:50, 550738, BD Biosciences) (Eberle *et al*, 2011). Whole cell DNA from isolated cones and rods were analysed by next generation sequencing (NGS).

### Intraocular AAV vector injections

In mice, subretinal injections were performed as previously described (Mühlfriedel *et al*, 2013) and intravitreal injections were done with slight modifications. In brief, mice were anaesthetised by intraperitoneal injections of ketamine (0.1 mg/g) and xylazine (0.02 mg/g). Tropicamide eye drops were applied to dilate the pupils (Mydriadicum Stulln). Injections were performed either free hand or with the UMP3T-1 Microinjection Syringe Pump using the Nanofil Sub-Microliter Injection System (World Precision

Instruments) equipped with a 34-gauge bevelled needle under an Opmi 1 FR pro surgical microscope (Carl Zeiss). For intravitreal injections, the needle bevel was turned down while penetrating the sclera, choroid and retina. Intravitreal positioning of the needle was confirmed using the surgical stereomicroscope. One microliter of rAAV vector solution was injected into the vitreous of the animals. Special care was taken to avoid damage of the lens.

For dog injections, animals were premedicated with subcutaneous or intramuscular acepromazine (0.02–0.025 mg/kg, Henry Schein Animal Health). Then anaesthesia was induced with intravenous propofol (4–6 mg/kg, PropoFol, Abbott Animal Health), the dogs were intubated, and anaesthesia maintained with isoflurane (Isoflo, Abbott Laboratories, between 2 and 3.5 % in a 1–2 l/min oxygen flow) via a rebreathing circle system. They received topical tropicamide solution (UPS 1%, Falcon Pharmaceuticals Ltd.) to induce mydriasis. Injections were performed under direct visualization using a microscope and a Machemer irrigating vitrectomy lens after routine aseptic preparation of the eye with betadine. All dogs received 200 µl of vector solution containing a total of 2E11 vg. The injection technique was performed as previously described (Gearhart et al, 2010; Mowat et al, 2014). Post-operatively, dogs received a subconjunctival injection of a steroid and antibiotic combination (2 mg methylprednisolone acetate, 0.1 mg dexamethasone and 1 mg gentamicin). Prednisone was given for 1 month at decreasing doses post-surgery starting at 1mg/kg once daily and the dogs were regularly examined for any signs of inflammation.

For non-human primate (NHP) injections, animals were sedated to effect NHP by intramuscular injection of ketamine (10 mg/kg) and dexmedetomidine (0.02 mg/kg) and placed in dorsal recumbence. Topical tropicamide solution and phenylephrine were administered for mydriasis. Following application of topical proparacaine for local anaesthesia, the eyelid margins were swabbed with undiluted 5% betadine solution and conjunctival fornixes were flushed with 0.5% betadine solution. A calliper was used to mark a spot 3.0 mm posterior to the limbus on the supratemporal bulbar conjunctiva, and the spot was swabbed with 5% betadine solution. A transparent adhesive drape was placed over the palpebral fissure, and an eyelid speculum was inserted. A 50 µl aqueous paracentesis was performed. A 31-gauge needle on an insulin syringe was inserted at the marked spot, through the sclera and advanced into the vitreous humour. The needle bevel was positioned to face the posterior axis of the globe and the contents delivered into the mid-vitreous by slowly depressing the syringe plunger. The needle was held in place for at least 2 min to lessen reflux of the injected material. When total injection volume exceeded 50 µl, they were split into two injections of equal volume to prevent elevated intraocular pressure.

## Scanning laser ophthalmoscopy

Fluorescent confocal scanning laser ophthalmoscopy (cSLO) imaging of the retina was performed in anaesthetised animals. Animals were anaesthetised and their pupils dilated as described above. Mouse retinas were examined using a modified Spectralis HRA + OCT system (Heidelberg Engineering) at 488 nm excitation and BP 550/49 emission as previously described (Schön et al, 2016). Unless otherwise stated, images were taken weekly for the 3-week

study duration, at the maximum detector sensitivity (107) in high-resolution mode with the scanner set to 30° field of view.

Dogs received topical tropicamide for mydriasis for ophthalmic examinations and retinal imaging. Regular ophthalmic examination and regular colour fundus images (RetCam II, Clarity Medical Systems, Pleasanton) were captured post-injection, either awake (using topical proparacaine for corneal anaesthesia) or after cSLO examination (under general anaesthesia) as previously described (Mowat et al, 2014). Wide-field blue-laser (488 nm) cSLO images were acquired using a Spectralis HRA + OCT at a sensitivity setting of 60–107 (images recorded at increments of 10: from 60 to 100 and 107; maximum setting to monitor eGFP expression. cSLO images were collected weekly from 4 weeks following intravitreal vector administration.

NHPs were sedated to effect, as above, and placed in sternal recumbence. Topical tropicamide and phenylephrine were administered for mydriasis. Wide-field blue-laser cSLO images were acquired from the central and peripheral retina using a Spectralis HRA + OCT to document distribution of eGFP expression at a sensitivity setting of 100. Images were collected weekly following vector administration for the 8-week study duration. Wide-field colour fundus images were also collected using a RetCam Shuttle (Natus Medical Incorporated), following instillation of topical proparacaine, during weeks 4, 6 and 8 following vector administration.

## Human retinal explant culture

Eyes from human donors were transferred into $CO_2$-independent medium (18045-054, Thermo Fisher) after surgical removal. 2–3 h after surgery, the retinas were cut into 6 pieces and transferred to cell culture inserts with 30 mm diameter and 0.40 µm pore size (PICMORG50, Millipore). Photoreceptor side of the retina was facing to the membrane. The retinal explants were cultured in Neurobasal A medium (10888022, Thermo Fisher) supplemented with 2mM L-Glutamine (25030024, Thermo Fisher), B27 supplement (17504044, Thermo Fisher) and Antibiotic-Antimyotic (15240062, Thermo Fisher) at 37°C, 5% $CO_2$. 70% of cell culture medium was replaced every day. 40 µl of AAV suspension was applied at the first day of culture to the ganglion cell layer (GCL) side of the tissue. eGFP fluorescence was recorded using an EVOS FL cell imaging system (Thermo Fisher) at 2× and 10× magnification. After an incubation period of 9–10 days, until which no severe tissue compromise was detected, cultures were fixed with 4% paraformaldehyde (PFA).

## Subcellular fractionation

HeLa cells, cultured in DMEM (10% FBS 1% Pen/Strep), were plated in 6-well plates. 24 h later, cells were counted and transduced with vector particles at a particle per cell ratio of 1,000. 24 h later, cells were harvested by extensive trypsin treatment and PBS washing steps to remove any membrane bound vector particles. Cells were counted and $3–5 \times 10^5$ cells were used for subcellular fractionation, while $10^5$ cells were analysed by flow cytometry (CytoFlex S platform, Beckman Coulter) to determine transduction efficiency. Subcellular fractionation was performed as previously described (Rossi et al, 2019). Membrane, cytosolic and nuclear

fractions were collected. Purity of fractions was confirmed by Western blot using anti-Rab 5 (1:100, sc46692, Santa Cruz), anti-Tubulin (1:5,000, T5198, Sigma Aldrich), anti-Lamin B1 (1:5,000, 16048, Abcam) and anti-Calreticulin (1:100, PA3-900, Affinity BioReagents) antibodies. Fractions were subjected to DNA isolation (Qiagen, DNeasy Tissue kit) followed by qPCR analysis (FastStart essential DNA green master reagent, Roche) using CMV promoter-specific primers on the LightCycler 96 real-time PCR system (Roche). The specificity of target DNA amplification was confirmed by melting-curve analysis. All samples were run in technical duplicates.

### Heparin affinity assay

A 1 ml HiTrap heparin affinity column (Amersham) was equilibrated with PBS/1 mM $MgCl_2$/2. 5 mM KCl. Vector preparations of AAV2, AAV2.NN and AAV2.GL purified by iodixanol gradient were diluted in PBS/1 mM $MgCl_2$/2.5 mM KCl and loaded on the column. Flow-through as well as $4 \times 5$ ml wash fractions were collected. In order to determine the affinity, elution was performed in 5 ml-steps using increasing ionic strength (PBS/1 mM $MgCl_2$/2.5 mM KCl plus 100 mM NaCl up to PBS/1 mM $MgCl_2$/2.5 mM KCl plus 1 M NaCl). DNA of samples was isolated (DNeasy Tissue kit, Qiagen) and analysed by qPCR on LightCylcer 96 real-time PCR system (Roche) using transgene-specific primers. The specificity of target DNA amplification was confirmed by melting-curve analysis. As a second independent measure, a native dot blot was performed using the capsid antibody A20 (1:50) (Wistuba et al, 1997) for probing (kindly provided by Martin Müller, DKFZ, Heidelberg, Germany).

### Neutralisation assay

HeLa cells were seeded in a 12-well plate 1 day prior to the neutralisation assay. Human serum was diluted in DMEM medium containing 10% FBS and 1% Pen/Strep. Vector preparations of AAV2, AAV2.NN and AAV2.GL were added to the serum dilution. As negative control, vector preparations were diluted in medium without human serum. Following incubation for 1 h at RT, medium in the 12-well plate was exchanged by the rAAV vector-serum dilution samples. Number of transgene-expressing cells was determined by flow cytometry 48 h post-transduction.

### Thermal stability assay

A 96-well qPCR plate was loaded with 2E9 vector particles/well diluted in PBS and subjected to a temperature gradient using the LightCycler 96 System (Roche Life Science) as previously described (Rossi et al, 2019). Subsequently, samples were diluted in PBS and transferred to a nitrocellulose membrane using a vacuum blotter. Membrane was probed with B1 (1:5,000, 16048, Abcam) antibody recognising the C'-terminus of all AAV2 capsid proteins and with A20 (1:50) antibody (Wistuba et al, 1997) binding to intact capsids. B1 and A20 were kindly provided by Martin Müller (DKFZ, Heidelberg, Germany). As secondary antibody, a horseradish peroxidase-conjugated anti-mouse antibody (1:10,000, Sigma-Aldrich) was applied. Finally, the membranes were treated with an enhanced chemiluminescence reagent (West Dura, Pierce) and analysed by FusionFX device (Peqlab).

### Quantification of AAV-transgene expression

RNA was isolated from fresh retinal tissue using the RNeasy Plus Mini Kit (74136, Qiagen) according to manufacturer instructions or from fixed tissue slices using an adapted protocol from the Allprep DNA/RNA FFPE kit (80234, Qiagen). Briefly, tissue slices (10–14 μm) were scraped with a sterile scalpel from 2 glass slides and collected in an Eppendorf tube. For mouse, dog and NHP material 10, 6 and 6, tissue slices were pooled, respectively. Subsequently, the material was deparaffinised by dissolving the scrapings in PKD buffer followed by a proteinase K digestion at 56°C for 15 min and then at 80°C for 15 min. The solution was finally chilled for 3 min on ice and centrifuged at $> 17,000$ g. The supernatant was then used for RNA extraction following the manufacturer instructions. Extracted RNA from both sources was digested with the RQ1 RNase-free DNase (M6101, Promega) to eliminate DNA contaminants and 100–200 ng were reverse transcribed using Superscript IV (18090010, Thermo Fisher). Gene specific primers (eGFP for: 5'-CGACCACTAC CAGCAGAACAC-3'; eGFP rev: 5'-TTCTCGTTGGGGTCTTTGCTCA G-3', mALAS for: 5'-TCGCCGATGCCCATTCTTATC-3', mALAS rev: 5'-GGCCCCAACTTCCATCATCT-3', ITR2 for: 5'-GGAACCCCTAGT GATGGAGTT-3', ITR2 rev: 5'-CGGCCTCAGTGAGCGA-3') were used to detect expression via RT–qPCR using the PowerUp™ SYBR™ Green master mix (A25742, Thermo Fisher). eGFP expression was normalised to ITR2 amplicons to account for variability in AAV genome quantities.

### Electroretinography

Mice were anaesthetised and tropicamide eye drops were applied for pupil dilation (Mydriadicum Stulln, Pharma Stulln GmbH). Full-field electroretinography (ERG) responses were recorded using a Celeris apparatus (Diagnosys LLC). Light guide electrodes which are embedded in the stimulators were placed on each eye. After a 5-min light adaptation step with 9 $cd/m^2$, sequential photopic responses were recorded for single-flash steps 1, 3 and 10 $cd.s/m^2$ with constant 9 $cd/m^2$ background illumination. Flicker recordings were performed for 1, 5 and 10 Hz at 3 $cd.s/m^2$ with constant 9 $cd/m^2$ background illumination.

### Immunohistochemical analysis

After cervical dislocation, mouse eyes were enucleated, fixed in 4% PFA/PBS for 1 h at RT followed by overnight incubation in 30% sucrose/PBS and embedded in optimal cutting temperature (OCT) reagent. For dog and NHP, after humane euthanasia (overdose of pentobarbitone), eyes were enucleated, slits made into the vitreal cavity at the pars plana region and the entire globes were immersed in 4% PFA at 4°C for 3.5 h for the dogs and 3.25 h for the NHP. The globes were then processed as previously described (Mowat et al, 2014). For human explants, the tissues were fixed in 4% PFA for 2 h at RT followed by overnight incubation in 30% sucrose/PBS and subsequent embedding in OCT. Extra care was taken to prevent detachment of the explant from the underlying filter. The primary antibodies used were anti-GFP chicken polyclonal (1:1,000; GFP-1020, Aves Labs), anti-GFP rabbit polyclonal (1:500; PABG1, Chromotek), anti-Recoverin rabbit polyclonal (1:1,000; AB5585, Merck), anti-Rhodopsin mouse monoclonal IgG1 clone ID4 (1:500; P21940,

Thermo Fisher), anti-Opsin (red/green) rabbit polyclonal (1:500; AB5405, Merck), custom anti-CNGA3 rabbit polyclonal (1:500) (Biel *et al*, 1999; Michalakis *et al*, 2005) and Lectin-PNA-A594 conjugate (1:500, L32459, Invitrogen). The secondary antibodies used were goat anti-chicken Alexa 488 (1:1,000, A32931, Thermo Fisher), goat anti-rabbit Alexa 488 (1:1,000, A11008, Thermo Fisher), donkey anti-rabbit Alexa 555 (1:500, A31572, Thermo Fisher) and goat anti-mouse Alexa 647 (1:500, A21236, Thermo Fisher). Cell nuclei for all sections were stained either with Hoechst-33342 or DAPI (1:2,000; D1306, Thermo Fisher). Confocal images were collected using a Leica SP8 confocal laser scanning microscope. The quantification of eGFP$^+$ photoreceptors in the ONL of NHP sections was done using ImageJ. A rectangle with fixed 5 mm$^2$ area was drawn over the ONL of xy images and nuclei which co-localised with eGFP signal were manually counted. Due to the absence of biological replicates for NHP, technical replicates were performed from different sections ($n \geq 3$).

## Statistics

Animals of the same age were genotyped and randomly assigned to the experimental groups irrespective of sex. The investigator was blinded during the *in vivo* analyses of the mouse studies. Data were collected from biological repeats $n > 3$; where technical repeats are shown instead of biological repeats this is clearly stated. Graphs and statistical analyses were performed using Prism 9 (GraphPad, San Diego). The results are presented as mean ± SEM. Unpaired Student's *t*-test was used to compare two parametric sample populations. For more than two populations, a one-way ANOVA and Holm–Sidak multiple comparisons test was performed. For grouped datasets ordinary two-way ANOVA or multiple *t*-tests using the Holm–Sidak method was used, assuming sample populations have the same scatter (SD). The significance level of $\alpha = 0.05$ was accepted ($P < 0.05$ *, $< 0.01$ **, $< 0.001$ ***).

# Data availability

This study includes no data deposited in external repositories.

**Expanded View** for this article is available online.

# Acknowledgements

We would like to thank Dr Daniela Meilinger, Kerstin Skokann, Anna Winkler, Ksenia Vershinina and Fred Koch from LMU for their technical assistance. We also thank Laura Escalona Espinosa and Hanna Janicki from the Centre for Molecular Medicine Cologne as well as Elke Barczak and Nicki Lenort from the Institute of Experimental Haematology, Hannover Medical School for their technical assistance. We thank Margarete Odenthal (University of Cologne) and Liang Zhang (Centre for Molecular Medicine Cologne) for the NGS. We also thank Janice Querubin (Michigan State University) for assisting with the large animal studies. Finally, we thank Jude Samulski (University of North Carolina at Chapel Hill, NC) for providing plasmid pXX6. The graphical abstract was created with BioRender.com. The molecular graphics were performed with UCSF Chimera, developed by the Resource for Biocomputing, Visualisation and Informatics at the University of California, San Francisco, with support from NIH P41-GM103311. This work was supported by grants from the Deutsche Forschungsemsachaft

(DFG) to S.M. (SPP2127, project MI 1238/4-1), DFG-funded cluster of excellence Centre for Integrated Protein Science Munich (CIPSM) to S.M. and M.B., Centre for Molecular Medicine Cologne (CMMC) to H.B., BMBF and MWK Lower Saxony-funded Professorinnenprogramm Niedersachsen to H.B., and DFG-funded cluster of excellence REBIRTH to H.B. The dog studies were funded by Myers-Dunlap Endowment to SP-J. Open access funding enabled and organized by ProjektDEAL.

# Author contributions

Project design, data analysis and experiment supervision: HB and SM; Mouse studies, large animal study sample analysis, human explant culture performance and analysis: MP; Library screens and mouse and human explant culture experiments: CS and SM; Sublibrary production: NM and HB; *In vitro* characterisation experiments: AR, NM and HB; Large animal study design and supervision: SMP-J; Large animal studies: LMO; NHP study design and performance: RFB and JTB; Tissue for human retinal explant experiments: MJG, JS and SGP; Mouse experiments: SB and JEW; Capsid modelling: MP and JB; *Cnga3*$^{-/-}$ mice and mouse study design: MB; Manuscript writing: MP, HB and SM; All authors contributed to comments and editing.

# Conflict of interest

S.M. and M.B. are co-founders of the gene therapy company ViGeneron GmbH who owns the rights on the related patent application WO/2019/076856 covering the novel rAAV capsids. All other authors declare no conflict of interest.

---

## The paper explained

### Problem

Loss of vision is one of the most severe handicaps with high socioeconomic importance. More than 5 million patients suffer from inherited retinal dystrophies, making them the leading cause of blindness in young individuals. A first gene therapy for one specific form of inherited blindness is now clinically available. However, many other conditions caused by mutations in hundreds of other genes remain without treatment. The most promising treatment concept is to supplement a healthy copy of the mutated gene to the affected cells of the retina; this requires the use of a viral vector. Currently, the standard clinical practice for delivering a therapeutic vector is under the patient's retina, which is effective but harbours risks of collateral damage and only treats a small portion of the affected retina. This work addresses the need for better viral vectors that can target the light-sensing cells of our retina more efficiently, using a minimally invasive route of administration.

### Results

We engineered two novel adeno-associated virus vectors with optimised properties for ocular gene therapy. We show that these novel vectors have a high efficacy for targeting the light-sensing cells in three animal models using a simple intraocular delivery. These vectors can also target human cells in culture. For one of these novel vectors, we provide first proof-of-concept data of vision restoration after gene supplementation in a mouse model of inherited total colour blindness.

### Impact

The use of these novel viral vectors could help improve patient outcome both in terms of efficiency and minimising the collateral damage evoked during administration. This could make gene therapies more successful and accessible to a wider patient pool, as the administration method is simpler and could be performed by any trained ophthalmologist.

## For more information

http://www.klinikum.uni-muenchen.de/Augenklinik-und-Poliklinik/de/forschung/gentherapie/english_version/index.html

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
