## [Review Process File · EMBO Molecular Medicine]

Novel AAV capsids for intravitreal gene therapy of photoreceptor disorders

Marina Pavlou, Christian Schön, Laurence Occelli, Axel Rossi, Nadja Meumann, Ryan Boyd, Joshua Bartoe, Jakob Siedlecki, Maximilian Gerhardt, Sabrina Babutzka, Jacqueline Bogedein, Johanna Wagner, Siegfried Priglinger, Martin Biel, Simon Petersen-Jones, Hildegard Büning, and Stylianos Michalakis

DOI: 10.15252/emmm.202013392

Corresponding authors: Stylianos Michalakis (stylianos.michalakis@med.uni-muenchen.de) , Hildegard Büning (Buening.Hildegard@mh-hannover.de)

Review Timeline:

Submission Date:	3rd Sep 20
Editorial Decision:	30th Sep 20
Revision Received:	30th Nov 20
Editorial Decision:	12th Jan 21
Revision Received:	14th Jan 21
Accepted:	15th Jan 21

Editor: Zeljko Durdevic

Transaction Report:

30th Sep 2020

Dear Prof. Michalakis,

Thank you for the submission of your manuscript to EMBO Molecular Medicine. We have now heard back from the three referees who agreed to evaluate your manuscript. As you will see from the reports below, while referee #1 and #3 are overall supporting publication of your work, referee #2 highlights the interest of the study but also raises a number of concerns particularly regarding the conceptual flaws and experimental design. While these critics can be addressed in writing, a cross-commenting exercise made it clear that the experiments in mice and non-human primates are not supporting the conclusions and would have to be repeated to benchmark the new AAV capsid variants to AAV2 and AAV2.7m8 as suggested by the referees #1 and #2. We would strongly suggest that you take the time and redo these experiments in a major revision of the current manuscript to strengthen the main conclusions of the study. New experiments in mice should include more mice to increase statistical significance and a photoreceptor specific promoter to evaluate the photoreceptor-specific transduction efficacy of the new vectors. In non-human primates, experiments with the parental capsids AAV2 and AAV2.7m8 should be included and the performance of the new capsids should be compared to that of the parental ones.

Nevertheless, given the overall positive evaluation of the manuscript by the referees #1 and #3 we would be willing to consider your manuscript without and additional experimentation if you tone down the conclusions and discuss the limitations of the study in regard to the experiments with mice and non-human primates. Addressing the reviewers' concerns in full, experimentally or in writing, will be necessary for consideration of your manuscript in our journal, and acceptance of the manuscript will entail a second round of review. Please note that EMBO Molecular Medicine encourages a single round of revision only and therefore, acceptance or rejection of the manuscript will depend on the completeness of your responses included in the next, final version of the manuscript. For this reason, and to save you from any frustrations in the end, I would strongly advise against returning an incomplete revision.

We would welcome the submission of a revised version within three to six months for further consideration. However, we realize that the current situation is exceptional on the account of the COVID-19/SARS-CoV-2 pandemic. Please let us know if you require longer to complete the revision.

I look forward to receiving your revised manuscript.

Yours sincerely,

Zeljko Durdevic

***** Reviewer's comments *****

Referee #1 (Comments on Novelty/Model System for Author):

Mice were used to develop the novel serotypes. I did not expect great results for the target NHP, but it seemed to work well. The novel AAVs were then applied to dog, NHP and human. Dog and NHP were good choices, but human retina was in a dish and therefore did not test the ability of these capsids to penetrate the retina.

Referee #1 (Remarks for Author):

Summary

This is an excellent study by two groups in Germany and the United States. Their goal was to develop novel engineered AAVs to permit intravitreal rather than subretinal injection to target photoreceptors. They made two engineered AAVs by IV injection in mice followed by isolation of DNA from target cells 24 hours later. At the third round they sorted rods and cones by FACS screening and identified AAV variants that reached rods and cones with counterselection for variants that bind to heparin. They were made up into self-complementary AAV vectors and injected intravitreally with CMV promoter into C57BL6 mice. Expression in photoreceptors was evaluated with cSLO imaging. Histology showed that two vectors, GL and NN gave the brightest expression in retina. They compared both to AAV2 and 7m8 in a vector of CMV-eGFP evaluated with cSLO. Strongest expression was in 7m8, GL and NN eyes. AAV2 vector generated expression mostly in RGCs but a little bit in INL and ONL. GL and NN was stronger than 7m8 and colocalized with recoverin in ~5% (my guess) of innersegments (Figure 1D). GL and NN proved less susceptible to inactivating antibodies in human serum. They then showed good photoreceptor expression after intravitreal injection in dogs (2E11 vg in AAV2, GL or NN CMV-eGFP). and NHPs. They report that both serotypes are also less sensitive to neutralizing antibodies in human serum. They evaluated the ability of the two serotypes to express in human photoreceptors by exposing human retinal explants to vector of CMV-eGFP. Finally, as a proof of principle they used the GL serotype to insert Cnga3 into Cnga3^{-/-} mouse and got good Cnga3 expression in cones.

Overall evaluation

All of these procedures were well tailored to the goal of the study and all were well carried out and well described.

Concerns

I have several concerns about the manuscript, mostly confined to missing control conditions and failure to temper some conclusions.

It was unclear how many dogs were injected, was it three groups of 5 dogs each? Data was shown for only one dog from each group and the details of GFP expression in individual photoreceptors could not be determined from Figure 2. This issue is important because the capsid development here is meant to replace subretinal injections, the traditional approach to photoreceptors, and the intravitreal injections should have been compared to subretinal injections.

Page 11 says "Inflammation was marked in few (sic) eyes treated with all three vectors and therefore those animals were terminated early". Inflammation is an important issue, since intravitreal injection produces substantially more inflammation than subretinal injection. The reader needs clear

description of this result, particularly a comparison of the novel capsids with AAV2.

On page 13 expression was described as "overlapping but not limited to nerve fibers and major blood vessels". What does that mean?

The NHP studies were seriously flawed as only NN and GL were examined, unlike the rest of the paper that involved comparisons of these with either AAV2 alone or AAV2 and 7m8. Three eyes were injected with GL and one with NN. The discussion reports "improved penetrance when compared qualitatively to previous reports (Dalkara 2013)". Dalkara showed only a single NHP, but without a direct comparison to 7m8 in this paper this conclusion seems unjustified.

GL and NN were used to insert GFP into human photoreceptors in an in vitro prep. They found much better expression in photoreceptors than in the NHP study, but vector presumably diffused around the retina piece to both vitread and sclerad sides.

The authors also demonstrated restoration of an ERG response after using intravitreal GL to insert Cnga3 into Cnga3^{-/-} mice. The value of this experiment would have been greater for many readers if ERGs had also been shown for wild type mice. Showing simply that Cnga3^{-/-} mice have no ERG and those treated have some ERG is not highly informative.

Referee #2 (Comments on Novelty/Model System for Author):

There are many logical flaws in the design of the experiments presented in this manuscript. The ultimate aims of the experiments are not logically linked to the methods used for the conception of the viral vectors. The validation however has been done in large animal preclinical models as well as mice which are appropriate for a gene therapy program with downstream clinical application.

Referee #2 (Remarks for Author):

The manuscript by Pavlou et al. describes experiments done to create an AAV vector capable of photoreceptor transduction upon intravitreal injection using in vivo directed evolution bioengineering approach. Strangely, the authors use intravenous administration in mice to select for an AAV variant to be used intravitreally in humans. Although the dataset testing the obtained variants is extensive and technically sound; the manuscript contains logical flaws and erroneous assumptions particularly in the conception of the viral vector. It further contains several misleading statements in its description of the state of the art in retinal gene therapy which shows the authors' limited knowledge of the field. See below for specific comments.

Conceptual shortcoming:

1. In nature, evolution selects for individuals with the most advantageous variations to overcome a given selection pressure. Directed evolution, which mimics and accelerates evolution in a laboratory setting is based on this same principal of selecting protein variants (enzymes, viral vectors, antibodies...) that outperform natural variants at a given task. The selection pressure used is logically specific to the desired outcome. Here, the authors aim to obtain AAV variants that can overcome physical barriers to retinal transduction from the vitreous. Instead of injecting animals in the vitreous and selecting variants that can reach the target cell (photoreceptors) as it was

previously done- the authors use intravenous injections of a viral library to select variants that can get into the retina via the blood retina barrier. What is the rationale behind this seemingly aberrant choice? This is a major shortcoming of the study which removes all compelling evidence that something useful for the task in hand was chosen. Regardless of this flaw the variants selected seem to have some interesting properties in mice. Their usefulness in the primate retina is unclear at this stage.

2. The authors state that the library was "counter- selected for capsid variants with strong binding ability for heparan sulphate proteoglycan (HSPG)". It is not clear why one would need to select variants aimed at gene delivery from the vitreous for their lack of HSPG binding. In fact, it has been shown that HSPG binding is actually important for intravitreally delivered AAVs as it promotes accumulation of intravitreally delivered AAVs at the vitreo-retinal interface for better access to the retina but weakly influences their tropism (Woodard et al., 2016). With this in mind it is counterproductive to eliminate heparin binding with the aim of gaining better access to the retina from the vitreous.

Technical shortcomings:

- It is unclear why the incubation period was reduced to 24 hours in order to increase selection pressure. Selection pressure in AAV screens mostly refers to the number of particles applied but here the kinetics of passage across the blood retina barrier seems to be an unrelated parameter to the desired fitness of the AAV to be selected.
- For the characterisation of novel capsids after intravitreal injection in mice with respect to parental serotype and benchmark 7m8- the authors use CMV promoter to encode GFP in a small number of animals. This comparison does not measure the ability of the vectors to transduce photoreceptors. Moreover, the significance of comparison to 7m8 is not shown. The small number of mice used, the high variability in expression (error bars in Figure 1F) do not support the statement that the new variants outperformed 7m8.

Erroneous statements and other minor comments:

- The abstract states "The unique in vivo selection procedure involved intravenous administration of AAV- libraries." In vivo selection involving intravenous administration of AAV libraries has been done several times over (Daverman et al., 2016; Chan et al., 2017; Challis et al., 2019; Ravindra et al., 2020). This study is thus not unique in this aspect.
- Page 4 introduction: the authors state "in order to target photoreceptors in the outer retina, the only effective administration route so far has been subretinal injections". This statement is erroneous. There are several published studies showing photoreceptor transduction in large animals from the vitreous (Dalkara et al., 2013; Byrne et al., 2020). Intravenous injections of AAV have also been reported to transduce deep retinal layers including photoreceptors (Byrne et al., 2015; Simpson et al., 2019). Moreover, suprachoroidal injections also have recently lead to photoreceptor transduction (Yue et al., 2020; Han et al., 2020).
- It is incorrect that subretinal injections lead to a limited transduction of the retina. Recent studies using rational design or biomining approaches have revealed AAV variants that can spread beyond the boundaries of the subretinal bleb leading to efficient transduction of large retinal zones (Khabou et al., 2018; Boye et al., 2020).
- The authors state "using AAV2.GL to deliver Cnga3 in a mouse model of achromatopsia (Cnga3^{-/-} (Biel, Seeliger et al., 1999)), we report a first proof-of-concept restoration of photoreceptor function by intravitreal gene therapy." This statement is erroneous as a large number of studies over the past 11 years have already reported restoration of photoreceptor function by intravitreal gene

therapy (see some examples here: Park et al., 2009; Dalkara et al., 2013, Byrne et al, 2014; Byrne et al., 2015; Du et al, 2015; Roddy et al., 2017).

- The authors state "Specifically, the loop extension caused by the peptide insert is directed inwards, while in case of AAV2.7m8 it points outwards." This statement and the following statement "however, the loop extensions of both novel variants appear to be oriented towards opposite directions as in AAV7m8." are not supported by data but are hypothetical based on in silico models. They should be rephrased to reflect this uncertainty as recent structural studies revealed that 7m8 variants loops are floppy and do not point in any given direction (Bennett et al., 2020).

Referee #3 (Comments on Novelty/Model System for Author):

1. Technical Quality (inc. statistical analysis):

The technical quality is extremely high. A whole battery of different in vitro and in vivo methods have been applied in high quality. The statistical analyses are appropriate; however, sample sizes are relatively low, which is acceptable when larger animal models (dogs and non-human primates) are used. The data is still plausible and sound.

2. Novelty:

There is a steady demand in improving AAV gene delivery. Especially for retinal gene transfer, the injection of AAV particles into the vitreous has many advantages compared to subretinal injections. The authors have adequately demonstrated this aspect. There are similar approaches published using AAV2 based peptide display library screenings. However, the two top hits are novel and functioning in mouse, dog, primate and human test systems. Therefore, the two de novo screened viral capsids are of course novel and of a high impact.

3. Medical impact:

Due to the efficacy demonstration in many model systems including human retinal explants, it is very likely that these two capsids will find their way into clinical applications. It is important to find efficient and safe AAV capsids that don't get neutralized when injected into patients.

4. Adequacy of model system:

The used model systems go beyond the state of the art as most studies are only done in mice with limited translational impact. The authors have chosen all important model systems, which are required in clinical trials towards a therapy. Based on their previous work on achromatopsia, they demonstrated that one of the new capsids improves gene therapy in a mouse model.

Referee #3 (Remarks for Author):

The work by Pavlou, Schoen et al. addresses two extremely important points of retinal gene transfer: first, the availability of efficient AAV capsids for gene delivery to photoreceptors and second, the site of administration, i.e. intravitreal injection, which is easier to perform and avoids further retinal damage. Here, the authors succeeded in screening two very promising capsid candidates, which they successfully tested in mice, dogs, non-human primates and human retinal explants. This battery of model systems is very convincing and it is very plausible that the screened capsids, although after further assessment, will be applied in clinical settings. The manuscript is extremely well structured, most sections are clearly written and concise. The discussion covers all

relevant aspects and is very comprehensive. Overall, the manuscript is of very high quality, it is innovative and very relevant for the vision restoration community.

Some minor comments:

- Page 6: It would be great to introduce in more detail the capsid sequence that a naïve reader can easily follow.
- Figure 1 (A). Are also in vivo cSLO images taken, which were adjusted and normalized to NN/CMV-EGFP settings? The overexposure of the recombinant AAV capsids masks all structural details of a retinal tissue.
- Page 11: Five beagles have been taken for the experiments and then "few" were excluded because of "inflammation". Could you please provide numbers and discuss whether or not the inflammation stems from the new capsids?
- Page 13 and 17: The term "after necropsy" is odd. Were the animals euthanized for this particular study? If so, the term "euthanize" appears more adequate.

EMM-2020-13392 Point-by-point response

We thank the editor and the reviewers for their insights and suggestions. We have carefully addressed each issue raised as follows:

Reviewer 1

- 1. It was unclear how many dogs were injected, was it three groups of 5 dogs each? Data was shown for only one dog from each group and the details of GFP expression in individual photoreceptors could not be determined from Figure 2. This issue is important because the capsid development here is meant to replace subretinal injections, the traditional approach to photoreceptors, and the intravitreal injections should have been compared to subretinal injections.**

We understand this point and provide additional information to clarify the details of the dog study. A total of five funduscopically normal beagle dogs from a colony of dogs maintained at Michigan State University were included in this intravitreal administration study. Their exact treatment is indicated in the table below (Table 1). Each novel capsid (AAV2.GL and AAV2.NN) was injected into n=3 eyes, whereas the control AAV2 was injected in n=2 eyes. As results tend to be consistent between dogs (in our experience) n=3 was considered adequate in the interests of the 3Rs principle. Please note that the AAV2.NS delivery was part of a study not addressed in this manuscript.

Dog	Viral construct – right eye	Viral construct – left eye
RSA 15-088	AAV2//scCMV-eGFP	AAV2.NS//scCMV-eGFP
RSA 15-090	AAV2//scCMV-eGFP	AAV2.NS//scCMV-eGFP
RSA 15-091	AAV2.GL//scCMV-eGFP	AAV2.NN//scCMV-eGFP
RSA 15-092	AAV2.GL//scCMV-eGFP	AAV2.NN//scCMV-eGFP
RSA 15-093	AAV2.GL//scCMV-eGFP	AAV2.NN//scCMV-eGFP

Table 1. List of dogs in this study

The data we show in Figure 3 are representative of the animals euthanized at 6 weeks post-injection and in order to provide better detail of individual photoreceptor transduction, we have added panels D and E. There we show a close-up of AAV2.GL and AAV2.NN-treated retinas, stained for eGFP, rhodopsin (rods) and M-opsin (cones) antibodies, and indicate single eGFP+ photoreceptors co-labelled with either rhodopsin or M-opsin.

As it was not possible for us to repeat any large animal study due to the ongoing pandemic, we have performed a pilot mouse study to show how subretinal administration of our novel capsids compares to intravitreal administration. We injected subretinally 3E9 vg of either AAV2.GL/scCMV-eGFP or AAV2.NN/scCMV-eGFP and monitored eGFP expression over 4 weeks (Fig. EV6). It was promising to see the reporter signal spreading over time and photoreceptors transduced in the retinal periphery i.e. outside the subretinal bleb. Nevertheless the transduction efficiency achieved via subretinal administration was more localised than that of intravitreal administration (Figure 1C). With this we hope to address any concerns on the efficiency that our novel vectors have when delivered through the vitreous.

- 2. Page 11 says "Inflammation was marked in few eyes treated with all three vectors and therefore those animals were terminated early". Inflammation is an important issue, since intravitreal injection produces substantially more inflammation than subretinal injection. The reader needs clear description of this result, particularly a comparison of the novel capsids with AAV2.**

As seen in the response to comment 1, a total of five dogs were injected in this study. Three dogs (RSA 15-088, RSA 15-091, RSA 15-093) were euthanized 6 weeks post-injection. Two dogs (RSA 15-090, RSA 15-092) were euthanized 4 weeks post-injection due to the development of ocular inflammation in both eyes. This was not associated with any particular vector as the four eyes of these

two dogs were injected with AAV2, AAV2.GL, AAV2.NN and AAV2.NS at equal viral vector doses (see Table 1 above). The degree of inflammation was similar between the two eyes of each dog. The inflammation events were therefore independent of the vector capsid used for injection. In a previous study of different AAV2 capsid variants delivered in the dog by intravitreal injection inflammation was also detected in some dogs and investigated in more detail. While a neutralizing antibody response to AAV was detected, a study of peripheral blood mononuclear cells (PMBC) showed a robust response to eGFP but not to the capsid (Boyd, Boye et al 2016). Immune response to the widely used reporter gene eGFP is a recognized issue (Stripecke et al 1999). To communicate the occurrence of inflammation in more detail to the readers, we have updated the text on page 9: "Intraocular inflammation developed at 2 and 3 weeks post injection in two animals. The inflammation was independent of the vector capsid used, as the affected eyes had been treated with AAV2, AAV2.GL or AAV2.NN. The two affected dogs were euthanized at 4 weeks post injection; the remaining three animals showed no evidence of inflammation and were euthanized at 6-weeks post injection." We also mentioned the relevant studies on eGFP immunogenicity in the discussion on page 17.

3. On page 13 expression was described as "overlapping but not limited to nerve fibres and major blood vessels". What does that mean?

We see how this phrase was not clear. The intended meaning was that the eGFP signal sometimes localised within nerve fibres and close to blood vessels, as seen via cSLO. However, this phrase was removed, as the key message was the widespread eGFP signal.

4. The NHP studies were seriously flawed as only NN and GL were examined, unlike the rest of the paper that involved comparisons of these with either AAV2 alone or AAV2 and 7m8. Three eyes were injected with GL and one with NN. The discussion reports "improved penetrance when compared qualitatively to previous reports (Dalkara 2013)". Dalkara showed only a single NHP, but without a direct comparison to 7m8 in this paper this conclusion seems unjustified.

We appreciate this comment and would like to emphasise that the comparison we reported in the discussion was purely qualitative based on published cSLO images by Dalkara et al 2013. We agree with the reviewer and removed this statement as we did not perform a side-by-side comparison but referred to published data.

5. GL and NN were used to insert GFP into human photoreceptors in an in vitro prep. They found much better expression in photoreceptors than in the NHP study, but vector presumably diffused around the retina piece to both vitreal and scleral sides.

This is indeed an important point to clarify. The retinal explant culture is an *ex vivo* system with known limitations which we declare in the text on page 15: "Indeed, retinal explants are missing relevant *in vivo* barriers such as the ILM and therefore cannot exclude that viral transduction in the human retina could differ." We do not claim a comparison between the *ex vivo* human explants and *in vivo* NHP study. We only emphasise the value of demonstrating transduction of human retinal cells, and in particular photoreceptors, excluding that species-specific factors limit the translational potential of the novel capsids.

6. The authors also demonstrated restoration of an ERG response after using intravitreal GL to insert Cnga3 into Cnga3^{-/-} mice. The value of this experiment would have been greater for many readers if ERGs had also been shown for wild type mice. Showing simply that Cnga3^{-/-} mice have no ERG and those treated have some ERG is not highly informative.

We agree with the reviewer. We updated Figure 6 panel C of photopic ERG B-wave quantification, which now includes the responses of wild type control mice.

Reviewer 2

- 1. In nature, evolution selects for individuals with the most advantageous variations to overcome a given selection pressure. Directed evolution, which mimics and accelerates evolution in a laboratory setting is based on this same principal of selecting protein variants (enzymes, viral vectors, antibodies...) that outperform natural variants at a given task. The selection pressure used is logically specific to the desired outcome. Here, the authors aim to obtain AAV variants that can overcome physical barriers to retinal transduction from the vitreous. Instead of injecting animals in the vitreous and selecting variants that can reach the target cell (photoreceptors) as it was previously done- the authors use intravenous injections of a viral library to select variants that can get into the retina via the blood retina barrier. What is the rationale behind this seemingly aberrant choice? This is a major shortcoming of the study which removes all compelling evidence that something useful for the task in hand was chosen. Regardless of this flaw the variants selected seem to have some interesting properties in mice. Their usefulness in the primate retina is unclear at this stage.***

The reviewer is right that the most obvious strategy to follow would have been to select our library via intravitreal administration. As mentioned by the reviewer this has already been done by other groups with various outcomes. We decided to use a different strategy in order to achieve a very high selection pressure by administering the AAV library via intravenous injection and recovering AAV genomes from retinal cells.

The high selection pressure can be broken down to the individual biological barriers that successful vector variants would have to overcome: i) the host immune system; ii) the systemic clearance; iii) the blood vessel endothelial cell barrier and retina-blood-barrier (RBB). Within the retina, vectors would need to iv) escape from the retinal blood vessels and diffuse into the retinal tissue. If entry from the choroidal blood vessels is assumed, then vectors should v) move through the Bruch's membrane, the retinal pigment epithelial cell barrier, the photoreceptor extracellular matrix and the outer limiting membrane to finally enter the photoreceptor cells. If the entry pathway is the photoreceptor outer segment, then vi) the connecting cilium would need to be overcome as well. If entry from the vitreal blood vessels is assumed then vectors would need to vii) penetrate through the inner limiting membrane, ganglion cell layer, the inner plexiform layer, the inner nuclear layer and the outer plexiform layer to finally enter the photoreceptors at their synaptic endings. Upon cell entry, the vectors also need to viii) traffic through and escape the endolysosomal vesicle system, uncoat and finally shuttle their genome through the nuclear membrane into the cell nucleus. We narrowed the time window for overcoming these biological barriers to 24 hours.

Therefore, the number of barriers that had to be overcome in order to reach the retina, as well as the number of interactions with proteins in the blood and potential off-target tissues such as the liver and spleen was considerable. Despite this, variants accumulated in retinal nuclei already 24 hours post library application. If we had been unsuccessful in detecting AAV variants under these conditions, we would have opted to change the application route; this however was not the case. The validation experiments outlined in our manuscript confirmed that this non-obvious approach was successful and allowed us to achieve the primary goal of obtaining novel AAV capsid variants with a high transduction efficiency of retinal cells, in particular of retinal photoreceptors, across species.

- 2. The authors state that the library was "counter- selected for capsid variants with strong binding ability for heparan sulphate proteoglycan (HSPG)". It is not clear why one would need to select variants aimed at gene delivery from the vitreous for their lack of HSPG binding. In fact, it has been shown that HSPG binding is actually important for intravitreally delivered AAVs as it promotes accumulation of intravitreally delivered AAVs at the vitreo-retinal interface for better access to the retina but weakly influences their tropism (Woodard et al., 2016). With this in mind it is counterproductive to eliminate heparin binding with the aim of gaining better access to the retina from the vitreous.***

By counter selecting for variants with wildtype-like HSPG binding the idea was to overcome excessive capturing of viral particles at the ILM; not to eliminate HSPG binding completely, as was done by Woodard et al. (2016). The capsid variants taken forward (AAV2.GL/AAV2.NN) actually maintained an HSPG binding ability, however at a lower level than the parental wildtype AAV2 capsid. One could speculate that this endowed the novel capsid variants a better balance of ILM accumulation and retinal penetrance. This is something we cover in the text on Page 13: “Capsid variants with strong or wildtype AAV2-like affinity for HSPG were excluded before *in vivo* screening by counter-selecting the library on a heparin affinity column. HSPG serves as natural primary receptor of AAV2 (Kern, Schmidt et al., 2003, Opie, Warrington et al., 2003) and is responsible for broad tropism and limited spreading of AAV2. Intriguingly, it appears that the ability to bind HSPG, albeit with a lower affinity than AAV2, is critical for retinal transduction through the vitreous and therefore fostered the selection of variants with respective features from our library, agreeing with previous reports (Woodard, Liang et al., 2016).”

- 3. It is unclear why the incubation period was reduced to 24 hours in order to increase selection pressure. Selection pressure in AAV screens mostly refers to the number of particles applied but here the kinetics of passage across the blood retina barrier seems to be an unrelated parameter to the desired fitness of the AAV to be selected.***

We agree with the reviewer that decreasing the number of particles applied is a common decision in case selection pressure is to be increased. However, fast kinetics of passage across barriers and fast infection kinetics for the target cell population is another important point to consider. By reducing the incubation period to 24 hours, we i) selected for variants that were less prone to be cleared; ii) restricted the time the virions were “visible” outside a cell, and iii) restricted the time allowed to target a receptor that mediates virus/vector uptake in an efficient and fast manner. The latter can be considered as basis for the fast onset kinetics for (eGFP) gene expression observed for our variants (please see Fig. 1A and new Fig. 2A). Of note, despite the short incubation time we were able to detect AAV variants in the nucleus of our target cells, which we think is crucial for selecting AAV capsid variants with fast intracellular trafficking.

- 4. For the characterisation of novel capsids after intravitreal injection in mice with respect to parental serotype and benchmark 7m8- the authors use CMV promoter to encode GFP in a small number of animals. This comparison does not measure the ability of the vectors to transduce photoreceptors. Moreover, the significance of comparison to 7m8 is not shown. The small number of mice used, the high variability in expression (error bars in Figure 1F) do not support the statement that the new variants outperformed 7m8.***

In order to better compare the ability of the vectors to transduce photoreceptors, we have performed an additional study in mice, where animals were intravitreally injected with either AAV2, AAV2.7m8, AAV2.GL or AAV2.NN carrying a gene expression cassette where eGFP was under control of a rhodopsin promoter instead of a CMV promoter. The results of this study were added in the results section on page 7 and summarised in a new figure 2. Briefly, we observed enhanced transduction and faster onset kinetics of the novel capsids, particularly AAV2.NN (Fig. 2A). At the 3 WPI time point when animals were euthanized, eGFP expression was significantly higher in retinas treated with AAV2.NN compared to all other variants (Fig. 2B-C), which is in line with the rod-favouring origin of the capsid. AAV2.GL performed at least as well as AAV2.7m8 and significantly better than AAV2 (Fig. 2B-C).

- 5. The abstract states "The unique in vivo selection procedure involved intravenous administration of AAV- libraries..." In vivo selection involving intravenous administration of AAV libraries has been done several times over (Daverman et al., 2016; Chan et al, 2017; Challis et al, 2019; Ravindra et al, 2020). This study is thus not unique in this aspect.***

To the best of our knowledge, intravenous library administration to select for retinal tropism has not been previously reported. The references mentioned by the reviewer are all studies where libraries were administered intravenously to investigate tropism in the brain and spinal cord, not the retina.

6. Page 4 introduction: the authors state "in order to target photoreceptors in the outer retina, the only effective administration route so far has been subretinal injections". This statement is erroneous. There are several published studies showing photoreceptor transduction in large animals from the vitreous (Dalkara et al., 2013; Byrne et al., 2020). Intravenous injections of AAV have also been reported to transduce deep retinal layers including photoreceptors (Byrne et al, 2015; Simpson et al, 2019). Moreover, suprachoroidal injections also have recently lead to photoreceptor transduction (Yue et al, 2020; Han et al, 2020).

We see how the wording was not correct at this point. Our intention was to reflect on the standard clinical application of the AAV2-based gene therapy Luxturna. In fact we reference the various other administration routes mentioned by the reviewer in our discussion on page 12. To clarify this point in our introduction as well, we have changed the sentence on page 3 to: "In order to target photoreceptors in the outer retina, the standard clinical administration route so far is subretinal..."

7. It is incorrect that subretinal injections lead to a limited transduction of the retina. Recent studies using rational design or biomining approaches have revealed AAV variants that can spread beyond the boundaries of the subretinal bleb leading to efficient transduction of large retinal zones (Khabou et al., 2018; Boye et al, 2020).

We thank the reviewer for pointing out this inconsistency and agree that in contrast to conventional AAV serotypes, a few engineered AAV variants (including ours – see Fig. EV5) can spread to different extends beyond the subretinal bleb boundaries. We have reworded our introduction section on pages 3-4 to include the relevant references and clarify our point: "With the exception of certain engineered capsids (Boye, Choudhury et al., 2020, Khabou, Garita-Hernandez et al., 2018), conventional AAV serotypes are unable to spread laterally and achieve only local transduction, i.e. cells outside the subretinal bleb area are not exposed to sufficient amounts of the vector." We have also updated our discussion on page 15 as follows: "Furthermore, we demonstrated the potency of the novel capsids when administered subretinally instead of intravitreally (Fig. EV5). Both novel capsids resulted in high eGFP expression outside the subretinal bleb area, indicating that similar to previously reported engineered rAAVs (Boye et al., 2020, Khabou et al., 2018) our AAV2.GL and AAV2.NN support lateral spreading. Nevertheless, as vector lateral spreading cannot compensate for all other risk factors associated with subretinal application, we focused on validating these two novel capsids for their intravitreal penetrance in larger mammals and human tissue as they are more clinically relevant models."

8. The authors state "using AAV2.GL to deliver Cnga3 in a mouse model of achromatopsia (Cnga3^{-/-}(Biel, Seeliger et al., 1999)), we report a first proof-of-concept restoration of photoreceptor function by intravitreal gene therapy." This statement is erroneous as a large number of studies over the past 11 years have already reported restoration of photoreceptor function by intravitreal gene therapy (see some examples here: Park et al., 2009; Dalkara et al., 2013, Byrne et al, 2014; Byrne et al., 2015; Du et al, 2015; Roddy et al., 2017).

We see how this statement was not clear. Our intention was to state that we provided a first validation of AAV2.GL in a gene supplementation study in mice. We therefore amended the sentence on page 4 which now reads: "Using the Cnga3^{-/-} mouse model of achromatopsia (Biel, Seeliger et al., 1999), we also report a first validation of AAV2.GL in a proof-of-concept study for restoring cone photoreceptor function after intravitreal gene supplementation therapy."

9. ***The authors state "Specifically, the loop extension caused by the peptide insert is directed inwards, while in case of AAV2.7m8 it points outwards." This statement and the following statement "however, the loop extensions of both novel variants appear to be oriented towards opposite directions as in AAV7m8." are not supported by data but are hypothetical based on in silico models. They should be rephrased to reflect this uncertainty as recent structural studies revealed that 7m8 variants loops are floppy and do not point in any given direction (Bennett et al., 2020).***

Indeed our discussion on how the peptide insertions impacted the structure of the capsid is based on *in silico* models, which we had already stated in the text and specifically indicated how these models were computed. We agree with the reviewer that the recent study by Bennett et al., 2020 is of relevance here, as it shows that peptide-extended loops might generally be flexible in 3D space. This of course could also be the case in our novel variants. We have further updated the comparison between our *in silico* models and the experimentally determined AAV2.7m8 structure, which in the meantime became publicly available (PDB 6U0R), in Expanded View Figure 4G-H. To better reflect the hypothetical nature of our discussion we amended the text on page 13 to: "Based on our *in silico* modelling, the loop extensions of both novel variants appear to be oriented in the opposite direction compared to AAV2.7m8. This however is subject to confirmation via structural analysis, as recent work indicated that such peptide-extended loops can be flexible in 3D space (Bennett et al., 2020)."

Reviewer 3

1. ***Page 6: It would be great to introduce in more detail the capsid sequence that a naïve reader can easily follow.***

We appreciate the feedback and have generated Expanded View Figure 1 to introduce the peptide-display diversification strategy and capsid sequence in more detail.

2. ***Figure 1 (A). Are also in vivo cSLO images taken, which were adjusted and normalized to NN/CMV-EGFP settings? The overexposure of the recombinant AAV capsids masks all structural details of a retinal tissue.***

We understand this request and provide an image panel to better visualise structural details of the retinal tissue. The fundus fluorescence images are from AAV2.7m8, AAV2.GL and AAV2.NN treated animals, obtained using a lower cSLO sensitivity 80 instead of 107 used in figure 1A. In figure 1, we opted for normalising the settings to the signal obtained from the AAV2-treated eyes, in order to better depict the difference in expression between the engineered capsids to the parental AAV2. Nevertheless we hope that these additional fluorescence images cover any concerns of underlying structural differences between the imaged retinae.

3. Page 11: Five beagles have been taken for the experiments and then "few" were excluded because of "inflammation". Could you please provide numbers and discuss whether or not the inflammation stems from the new capsids?

Indeed we needed to elaborate on the dog study, as was also pointed out by reviewer 1 (see responses to comments 1 & 2). We have included further information about the dog study in our results section, indicating that the inflammation was independent of the vector administered in each eye. The corresponding sentence on Page 9 now reads: "Intraocular inflammation developed at 2 and 3 weeks post injection in two animals. The inflammation was independent of the vector capsid used, as the affected eyes had been treated with AAV2, AAV2.GL or AAV2.NN. The two affected dogs were euthanized at 4 weeks post injection; the remaining three animals showed no evidence of inflammation and were euthanized at 6-weeks post injection."

4. Page 13 and 17: The term "after necropsy" is odd. Were the animals euthanized for this particular study? If so, the term "euthanize" appears more adequate.

We appreciate this being pointed out; indeed the accurate term for these studies is euthanasia and we have replace the term in the text on pages 13 and 17.

12th Jan 2021

Dear Prof. Michalakis,

Thank you for the submission of your revised manuscript to EMBO Molecular Medicine. As you will see from the reports below, while referee #1 and #3 are supporting publication of your work, referee #2 remains critical and raises a number of concerns particularly regarding the lack of novelty and insufficient conceptual advance. Given the continuous positive evaluation of the manuscript by the two referees I am pleased to inform you that we will be able to accept your manuscript pending the following final amendments:

- 1) Please address all the concerns raised by the referee #2. No new experiments are required.
- 2) Tables: Please add Table 1 to the main text file and place it on the end of the file together with its legend.
- 3) In the main manuscript file, please do the following:
 - Correct/answer the track changes suggested by our data editors by working from the attached/uploaded document.

***** Reviewer's comments *****

Referee #1 (Comments on Novelty/Model System for Author):

I feel that the manuscript was thoroughly reviewed and that the issues raised were not fatal to publication but largely required additional commentary from the authors. The authors were very responsive to the concerns raised by the reviewers, and added substantial content to the manuscript to respond to the issues raised. I am now convinced that the manuscript has now resolved the many issues raised by reviewers.

Referee #2 (Remarks for Author):

Based on the instructions to the reviewers provided by EMBO Mol Med, and after revisions provided

by the authors, I do not recommend this manuscript for publication in EMBO Mol Med. The major claim of this manuscript remains that new rAAV vectors expanding the clinical applicability of gene therapy for blinding human retinal disorders have been created. This major claim was not supported by the data presented in the earlier version and is not supported with the inclusion of new data. The large animal data presented show that doses incompatible with safety in canines were used leading to early termination of experiments and NHP data still only show a similar performance to previously published variants. The variants presented here are at best equal to previously published ones and have not been compared to any of the newer variants created specifically in primates (Byrne et al, 2020) making the claim of better clinical applicability false. Additional data in mouse comparisons have been provided. However the authors present data at 3 weeks post infection which is a too early time point for optimal performance of 7m8 against which they compare their variants making this an unjustified comparison. As indicated in previous publications on 7m8, full development of transgene expression from 7m8 in photoreceptors is 6-8 weeks post injection. If their claim of better performance of AAV2.NN compared to all other variants is solely based on time, I don't see how a few weeks would translate to better clinical applicability in retinal degenerative diseases which take decades to progress. Moreover, the justification of the authors on the inadequacy of their selection process in view of the desired application falls short of being satisfactory. They insist on how considerable the barriers were such as the host immune system; the systemic clearance; the blood vessel endothelial cell barrier and retina-blood-barrier (RBB) but none of these barriers are relevant for an AAV variant that is administered into the vitreous. Thus, the major claim that novel and better capsids in view of intravitreal retinal gene therapy have been created thanks to a 'unique' selection is neither novel nor convincing.

In response to the referee question 'are the claims appropriately discussed in the context of earlier literature?' and the depth of analysis: integration and analysis of the literature cited - my answer remains unsatisfactory. Where new sentences have been added here and there for corrections or clarity, new erroneous statements have appeared such as the one on page 17.

I thus recommend rejection based on lack of novelty, insufficient conceptual advance and specialist interest for what is the rest (rescue using the new AAV variants in the CNGA3^{-/-} mouse model of achromatopsia).

Referee #3 (Comments on Novelty/Model System for Author):

All my points have been addressed and are clarified.

Referee #3 (Remarks for Author):

I would like to thank the authors for thoroughly addressing all my points within the revised manuscript.

The authors performed the requested editorial changes.

Reviewer 1

(Comments on Novelty/Model System for Author)

I feel that the manuscript was thoroughly reviewed and that the issues raised were not fatal to publication but largely required additional commentary from the authors. The authors were very responsive to the concerns raised by the reviewers and added substantial content to the manuscript to respond to the issues raised. I am now convinced that the manuscript has now resolved the many issues raised by reviewers.

We would like to thank the reviewer for their comments and appreciate their contribution in improving our manuscript.

Reviewer 2

(Remarks for Author)

Based on the instructions to the reviewers provided by EMBO Mol Med, and after revisions provided by the authors, I do not recommend this manuscript for publication in EMBO Mol Med. The major claim of this manuscript remains that new rAAV vectors expanding the clinical applicability of gene therapy for blinding human retinal disorders have been created. This major claim was not supported by the data presented in the earlier version and is not supported with the inclusion of new data.

We respect the opinion of the reviewer, but we strongly believe that our data support the claims made in our manuscript.

The large animal data presented show that doses incompatible with safety in canines were used leading to early termination of experiments...

The doses used in this study are similar to those published in previous studies of intraocular AAV delivery in canines (Boyd, Boye et al 2016; Boyd, Sledge et al 2016). As stated in our previous response and manuscript, the inflammation detected in two animals of our dog study was independent of the vector capsid used, as the affected eyes had been treated with AAV2, AAV2.GL or AAV2.NN, and was likely in response to eGFP (Boyd, Boye et al 2016).

...and NHP data still only show a similar performance to previously published variants. The variants presented here are at best equal to previously published ones and have not been compared to any of the newer variants created specifically in primates (Byrne et al, 2020) making the claim of better clinical applicability false.

Indeed, our non-human-primate study did not include a side-by-side comparison of our novel AAV variants to recently published variants (Byrne et al, 2020), which is why we have not claimed superiority over other variants in primates. We do however provide evidence of superiority in mouse. We clearly state that our aim was to expand the clinical applicability of gene therapy for IRDs with novel tools, and that is what we have demonstrated to the satisfaction of the other two reviewers.

Additional data in mouse comparisons have been provided. However the authors present data at 3 weeks post infection which is a too early time point for optimal performance of 7m8 against which they compare their variants making this an unjustified comparison. As indicated in previous publications on 7m8, full development of transgene expression from 7m8 in photoreceptors is 6-8 weeks post injection. If their claim of better performance of AAV2.NN compared to all other variants is solely based on time, I don't see how a few weeks would translate to better clinical applicability in retinal degenerative diseases which take decades to progress.

As the reviewer states, our novel vectors have faster kinetics than 7m8, with strong transgene expression already at 3 weeks post injection, meaning they were trafficked and processed more efficiently than other AAV vectors. For gene expression cassettes with strong promoters like CMV or RHO, in our experience, steady state levels of gene expression are already reached at 3 weeks post injection. Moreover, it is important to consider that retinal degeneration manifests at various rates and progresses distinctly in each individual patient (Russell, Bennett et al., 2017). Therefore, one should not exclude that certain patients, particularly those with early disease-onset, could profit from novel gene therapy vectors with improved properties, such as those demonstrated for AAV2.GL and AAV2.NN.

Moreover, the justification of the authors on the inadequacy of their selection process in view of the desired application falls short of being satisfactory. They insist on how considerable the barriers were such as the host immune system; the systemic clearance; the blood vessel endothelial cell barrier and retina-blood-barrier (RBB) but none of these barriers are relevant for an AAV variant that is administered into the vitreous. Thus, the major claim that novel and better capsids in view of intravitreal retinal gene therapy have been created thanks to a 'unique' selection is neither novel nor convincing. In response to the referee question 'are the claims appropriately discussed in the context of earlier literature?' and the depth of analysis: integration and analysis of the literature cited - my answer remains unsatisfactory. Where new sentences have been added here and there for corrections or clarity, new erroneous statements have appeared such as the one on page 17.

We respect the opinion of the reviewer, yet we find that better tools are often the result of out-of-the-box thinking, which is why we chose a screening strategy that has not been previously used for AAV capsid evolution in the retina.

On page 17 we inserted the following sentence: “*Although ubiquitous promoters are known to outperform cell-specific promoters, we show that our novel capsids effectively transduce cones and rods after intravitreal administration using photoreceptor-specific promoters, such as mSWS (Fig.6) and hRho (Fig. 2), respectively.*” We do not see how this statement is erroneous. In particular, there is published literature supporting this statement (McClements & MacLaren, 2013), which we now reference for clarity.

I thus recommend rejection based on lack of novelty, insufficient conceptual advance and specialist interest for what is the rest (rescue using the new AAV variants in the CNGA3-/- mouse model of achromatopsia).

We have incorporated all the suggestions made by the reviewers and demonstrated how this work contributes to and expands the clinical toolbox for gene therapy by the introduction of these **novel** vectors. As such, we do believe our manuscript is of great interest to the readers of EMBO Molecular Medicine.

Reviewer 3

(Comments on Novelty/Model System for Author)

All my points have been addressed and are clarified.

We are glad that our response was satisfactory.

(Remarks for Author)

I would like to thank the authors for thoroughly addressing all my points within the revised manuscript.

We in turn also thank the reviewer for their constructive comments, leading to the improvement of our manuscript.

References:

McClements ME, MacLaren RE (2013) Gene therapy for retinal disease. *Transl Res* 161: 241-54
Russell S, Bennett J, Wellman JA, Chung DC, Yu ZF, Tillman A, Wittes J, Pappas J, Elci O, McCague S et al. (2017) Efficacy and safety of voretigene neparvovec (AAV2-hRPE65v2) in patients with RPE65-mediated inherited retinal dystrophy: a randomised, controlled, open-label, phase 3 trial. *Lancet* 390: 849-860

15th Jan 2021

Dear Prof. Michalakis,

We are pleased to inform you that your manuscript is accepted for publication.

Corresponding Author Name: Stylianos Michalakis / Hildegard Büning

Manuscript Number: EMM-2020-13392